# Regulation of mammalian 3D genome organization and histone H3K9 dimethylation by H3K9 methyltransferases

Kei Fukuda [1] [✉], Chikako Shimura[1], Hisashi Miura[2], Akie Tanigawa[2], Takehiro Suzuki[3], Naoshi Dohmae[3], Ichiro Hiratani [2] & Yoichi Shinkai [1] [✉]

Histone H3 lysine 9 dimethylation (H3K9me2) is a highly conserved silencing epigenetic mark. Chromatin marked with H3K9me2 forms large domains in mammalian cells and overlaps well with lamina-associated domains and the B compartment defined by Hi-C. However, the role of H3K9me2 in 3-dimensional (3D) genome organization remains unclear. Here, we investigated genome-wide H3K9me2 distribution, transcriptome, and 3D genome organization in mouse embryonic stem cells following the inhibition or depletion of H3K9 methyltransferases (MTases): G9a, GLP, SETDB1, SUV39H1, and SUV39H2. We show that H3K9me2 is regulated by all five MTases; however, H3K9me2 and transcription in the A and B compartments are regulated by different MTases. H3K9me2 in the A compartments is primarily regulated by G9a/GLP and SETDB1, while H3K9me2 in the B compartments is regulated by all five MTases. Furthermore, decreased H3K9me2 correlates with changes to more active compartmental state that accompanied transcriptional activation. Thus, H3K9me2 contributes to inactive compartment setting.

[1] Cellular Memory Laboratory, RIKEN Cluster for Pioneering Research, Wako, Japan. [2] Laboratory for Developmental Epigenetics, RIKEN Center for Biosystems Dynamics Research, Kobe, Japan. [3] Biomolecular Characterization Unit, Technology Platform Division, RIKEN Center for Sustainable Resource Science, Wako, Japan. [✉]email: kei.fukuda@riken.jp; yshinkai@riken.jp

Post-translational modifications of histone proteins regulate chromatin compaction and mediate epigenetic transcriptional regulation. The methylation of histone tails is a fundamental event in epigenetic signaling. In eukaryotes ranging from the fission yeast Schizosaccharomyces pombe to humans, repeat-rich constitutive heterochromatin is marked by H3K9me2 or H3K9me3[1–3]. These modifications are catalyzed by a family of SET domain-containing lysine methyltransferases, of which five are present in mammals. SETDB1, with the related enzymes SUV39H1 and SUV39H2, contribute to the formation of H3K9me3[1,4], whereas GLP and G9a (also called EHMT1 and EHMT2, respectively) regulate H3K9me1 and H3K9me2 formation, respectively[5–7].

In mouse embryonic stem cells (mESCs), H3K9me3 is enriched in retroelements and pericentromeric satellite repeats that are mediated by SETDB1 and SUV39H1/2, respectively[8–10]. SUV39Hs are also involved in H3K9me3 modification in intact retroelements and LINE1[11]. Unlike H3K9me3, H3K9me2 forms megabase (Mb)-scale domains that comprise approximately half of the genome in both pluripotent and differentiated cells[9,12,13]. Endogenous G9a and GLP mostly exist as G9a/GLP heterodimers; this complex is the functional form for global H3K9 methylation in vivo[5,7].

Although H3K9me2 is primarily catalyzed by G9a/GLP in mESCs, not all H3K9me2 is diminished in either G9a or Glp knockout (KO) or G9a/Glp double knockout (DKO) mESCs[6,7], suggesting the involvement of other histone MTases in H3K9me2. However, the function and mechanism of G9a/GLP-independent H3K9me2 are currently unknown. As H3K9me2 is enriched in pericentromeric satellite repeats[6,14] and genomic regions with H3K9me3[15], SETDB1 and SUV39H1/2 may be involved in G9a/GLP-independent H3K9me2.

Recent studies using Hi-C technology have revealed that chromosomes are hierarchically folded at different levels in the nucleus, including chromatin loops, topologically associating domains, and nuclear compartments[16–19]. At the megabase scale, genomic regions are segregated into two subnuclear compartments that consist predominantly of either euchromatin (A compartment) or heterochromatin (B compartment). The B compartment is overlapped with lamina-associated domains[20,21], which interact with the nuclear lamina (NL) and highly overlap with H3K9me2 domains[13]. G9a-mediated H3K9me2 is required for LAD–NL interaction[22]. Based on the links among H3K9me2, LADs, and B compartments, both G9a/GLP-dependent and independent H3K9me2 could be involved in A/B compartment formation. Therefore, we analyzed H3K9me2 genome-wide profiles and 3D genome organization in Setdb1 KO and/or Suv39h1/2 DKO cells treated with a G9a/GLP-specific inhibitor, UNC0642[23], to identify the roles of each H3K9 MTase in H3K9me2 domain formation and to determine whether H3K9me2 is involved in nuclear compartment formation.

This study demonstrated that H3K9me2 is regulated by different sets of H3K9 MTases between the A and B compartments. Also, it was suggested that G9a/GLP-independent H3K9me2 has a role in efficient H3K9me2 recovery and transcriptional repression. Furthermore, this study also showed that decreasing H3K9me2 is linked to converting the genome to more active compartmental state, although the overall compartment pattern is maintained in H3K9me2-depleted cells. Thus, H3K9me2 contributes to repressive 3D chromatin state.

## Results

**Compartment-dependent regulation of H3K9me2 in mESCs.** It has been reported that H3K9me2 is enriched in LADs[13], which significantly overlaps with the B compartments in mESCs. Our H3K9me2 chromatin immunoprecipitation sequencing (ChIP-seq) in mESCs showed higher H3K9me2 enrichment in the B compartments (negative compartment score regions) than in the A compartments (positive compartment score regions) as expected; however, H3K9me2 was also present in the A compartments, with a relatively low compartment score (Fig. 1a). As SETDB1 and SUV39H1/2 have the potential to mediate H3K9me2 function, H3K9me2 ChIP-seq was performed in previously established Setdb1 conditional KO (cKO) mESCs[8] and in Suv39h1/2 DKO mESCs[10] (Supplementary Fig. 1a) to investigate the role of SETDB1 and SUV39H1/2 in H3K9me2 formation. H3K9me2 profiles in Setdb1 KO and Suv39h1/2 DKO mESCs were highly correlated with those of wild-type (WT) mESCs (Supplementary Fig. 1b). However, we found decreased H3K9me2 in the A compartments of Setdb1 KO mESCs (Fig. 1b) and in the B compartments of Suv39h1/2 DKO mESCs (Fig. 1c), respectively.

To obtain a more detailed view of the H3K9me2 profile, we identified H3K9me2 domains in mESCs using hiddenDomains, a program that uses a hidden Markov model to identify broad domains[24] (Fig. 1d). The H3K9me2 domains occupied 46.2% and 85.7% of the A and B compartments, respectively (Fig. 1e). The total length of H3K9me2 domains in the A compartments slightly decreased in Setdb1 KO mESCs (Fig. 1f), whereas the number of such domains was slightly increased (10,879 in WT and 12,022 in Setdb1 KO), suggesting that SETDB1 has a role in H3K9me2 domain expansion/spreading in the A compartments. It is difficult to further dissect SETDB1- and SUV39H1/2-dependent H3K9me2 in these KO mESC lines, owing to the dominant impact of G9a/GLP on H3K9me2 formation. The G9a/GLP catalytic inhibitor UNC0642 is useful for analyzing G9/GLP-independent H3K9me2[23]. Treatment with 0.5–2 μM UNC0642 for 3 days decreased H3K9me2 to a level comparable with that observed in G9a/GLP DKO mESCs (Supplementary Fig. 1c). Furthermore, H3K9me2 in G9a/GLP DKO mESCs and UNC0642-treated mESCs was enriched in pericentromeric satellite repeats, characterized as 4′,6-diamidino-2-phenylindole–(DAPI)-dense regions (Supplementary Fig. 1d, e), consistent with a previous report[14]. To assess genome-wide H3K9me2 states in more detail, we performed H3K9me2 ChIP-seq analysis in G9a/GLP DKO and UNC0642-treated mESCs. Compared with control WT mESCs, the experimental G9a/GLP DKO and UNC0642-treated mESCs formed relatively narrow H3K9me2 domains, with a high overlap between them (Supplementary Fig. 1f, g). Therefore, UNC0642 treatment mimics the H3K9me2 profile associated with the G9a/Glp DKO phenotype.

To investigate the roles of SETDB1 and SUV39H1/2 in G9a/GLP-independent H3K9me2, we performed H3K9me2 ChIP-seq from Setdb1 KO or Suv39h1/2 DKO mESCs treated with 2 μM UNC0642. Hierarchical clustering analysis of the H3K9me2 ChIP-seq data showed that the H3K9me2 profile of UNC0642-treated Setdb1 KO mESCs was largely distinct from that of G9a/GLP DKO mESCs (Fig. 1g). Consistent with untreated control mESCs, reductions of H3K9me2 in Setdb1 KO and Suv39h1/2 DKO mESCs treated with UNC0642 were mainly observed in the A and B compartments, respectively, with a significant decrease of H3K9me2 domains in the A compartments of UNC0642-treated Setdb1 KO mESCs (Fig. 1h, i, Supplementary Fig. 1h). This compartment-specific reduction of H3K9me2 in Setdb1 KO and Suv39h1/2 DKO mESCs was reproducible (Supplementary Fig. 1i). Approximately 80% of the G9a/GLP-independent H3K9me2 domains in the A compartments and ~20% of those in the B compartments were not present in the UNC0642-treated Setdb1 KO mESCs, whereas almost all the G9a/GLP-independent H3K9me2 domains were maintained in UNC0642-treated Suv39h1/2 DKO mESCs (Fig. 1j–l). Although SUV39H1/2

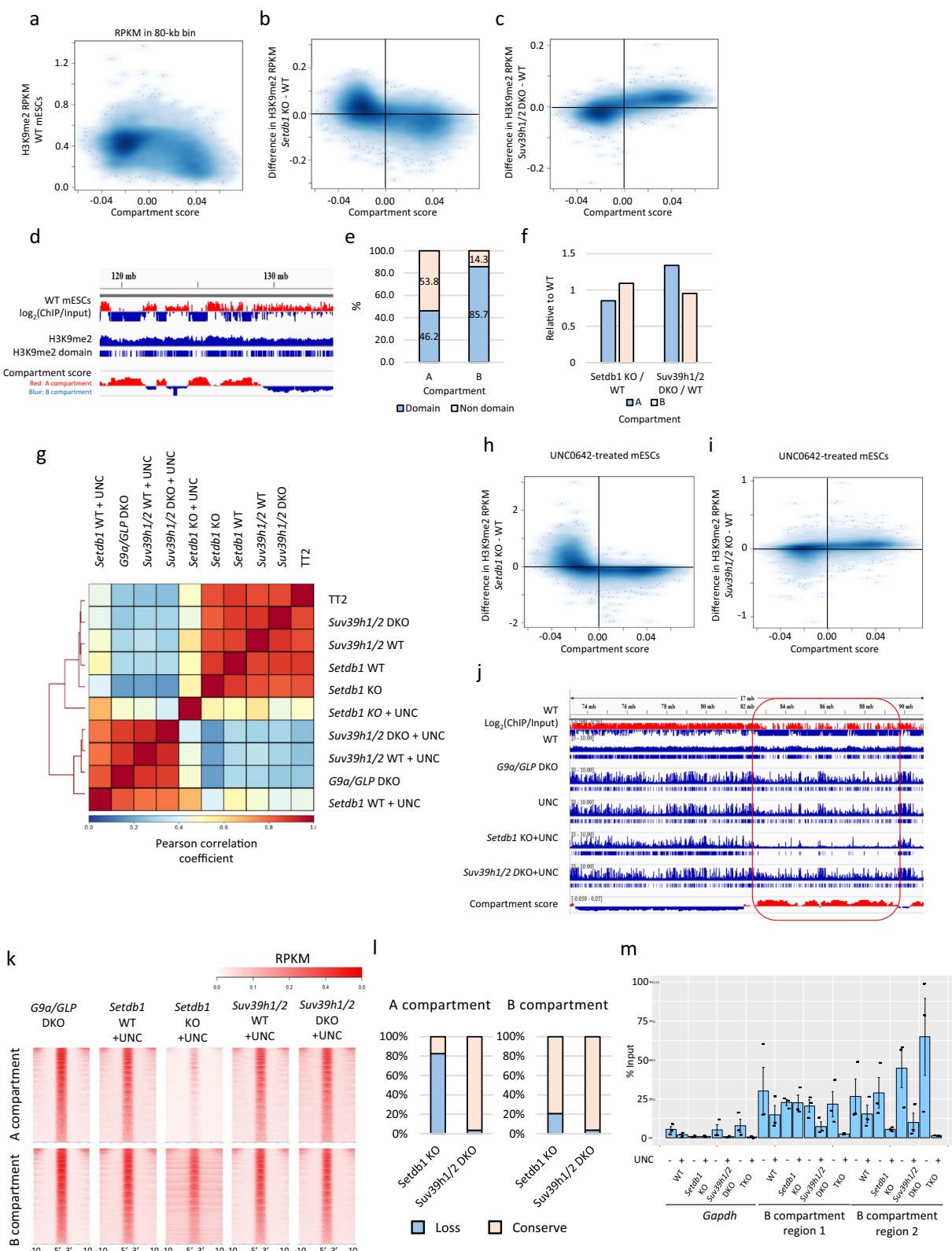

depletion did not have the same impact as SETDB1 depletion from the perspective of H3K9me2 ChIP-seq data, H3K9me2 immunofluorescence showed a noticeable decrease of H3K9me2 in DAPI-dense regions in UNC0642-treated *Suv39h1/2* DKO mESCs (Supplementary Fig. 1j). This finding suggests a crucial role of SUV39H1/2 in H3K9me2 in pericentromeric satellite repeats.

To determine whether H3K9me2 in the B compartment was mediated by SETDB1 and SUV39H1/2 redundantly, we established *Setdb1* and *Suv39h1/2* triple KO (TKO) mESCs using the CRISPR-Cas9 system (Supplementary Fig. 1k). The complete removal of *Suv39h1* exon 3, containing the SET domain, and the removal of SUV39H2 H398, which is essential for methyltransferase activity, was validated by RNA-seq in TKO mESCs[1]

**Fig. 1 Characterization of SETDB1 and SUV39H1/2-dependent H3K9me2 region in mESCs. a** Comparison between compartment score and H3K9me2 enrichment in 80-kb bin. RPKM of H3K9me2 from WT mESCs is negatively correlated with compartment score. Darker blue represents higher dot density. **b, c** Comparison between compartment score in WT mESCs and changes in H3K9me2 in *Setdb1* KO (**b**) or *Suv39h1/2* DKO (**c**) mESCs in 80-kb bins. Decreased H3K9me2 is observed in the A and B compartments in *Setdb1* KO and *Suv39h1/2* DKO mESCs, respectively. **d** A representative view of H3K9me2 domains in WT mESCs identified by *Hiddendomains*. Top panel, H3K9me2 ChIP-seq data from WT mESCs; middle panel, H3K9me2 domains; bottom panel, compartment score in 80-kb bins. **e** Fractions of H3K9me2 domains in the A or B compartments. Although H3K9me2 domains are more enriched in the B compartments than in the A compartments, H3K9me2 also occupies about half of the A compartments. **f** Length of H3K9me2 domains in each KO mESCs to WT mESCs. The length of H3K9me2 domains in the A compartments is slightly decreased in *Setdb1* KO mESCs, whereas that is increased in *Suv39h1/2* DKO mESCs. **g** A correlation matrix of H3K9me2 ChIP-seq data in mESCs. UNC0642-treated mESCs form a cluster distinct from untreated mESCs. Among UNC0642-treated mESCs, *Setdb1* KO mESCs show a different H3K9me2 profile. **h, i** A comparison between compartment score in WT mESCs and changes in H3K9me2 in *Setdb1* KO (**h**) or *Suv39h1/2* DKO (**i**) mESCs treated with UNC0642. **j** A representative view of H3K9me2 ChIP-seq from *G9a/GLP* DKO and UNC0642-treated mESCs (chr6:73,266,079-91,207,724). *G9a/GLP* DKO mESCs show an H3K9me2 profile similar to UNC0642-treated mESCs. A drastic loss of H3K9me2 domains in UNC0642-treated *Setdb1* KO mESCs in the A compartments (circled in red). **k** Heatmaps of H3K9me2 enrichment around G9a/GLP-independent H3K9me2 regions. The A compartment-specific H3K9me2 loss is observed in UNC0642-treated *Setdb1* KO mESCs. **l** Fractions of conserved or lost G9a/GLP-independent H3K9me2 in *Setdb1* KO or *Suv39h1/2* DKO mESCs treated with UNC0642. **m** H3K9me2 ChIP-qPCR analysis in regions shown in Fig. S1o. ChIP was performed 5 days after 4-OHT treatment in *Setdb1* KO mESCs and 7 days after 4-OHT treatment in *Setdb1/Suv39h1/2* TKO mESCs. H3K9me2 in the region 1 and 2 is completely lost in UNC0642-treated TKO mESCs. Data are mean ± SEM; *n* = 3.

(Supplementary Fig. 1k). The depletion of SUV39H1/H2 in TKO mESCs was further validated by western blotting (Supplementary Fig. 1l). We used 4-OHT-inducible conditional *Setdb1* KO mESCs as parental cells for the TKO mESC line to show that SETDB1 and H3K9me3 were significantly decreased at 7 days after 4-OHT treatment (Supplementary Fig. 1m). Furthermore, UNC0642 treatment of 4-OHT-treated TKO mESCs decreased H3K9me2 to an undetectable level by western blotting (Supplementary Fig. 1n). We confirmed the loss of H3K9me2 in the two regions where H3K9me2 was retained in both *Setdb1* KO and *Suv39h1/2* DKO mESCs treated with UNC0642 (Fig. 1m and Supplementary Fig. 1o). Therefore, G9a/GLP-independent H3K9me2 in the B compartments is mediated by both SETDB1 and SUV39H1/2.

**Compartment-dependent regulation of H3K9me2 in immortalized mouse embryonic fibroblasts (iMEFs).** G9a/GLP-independent H3K9me2 in the A and B compartments was mediated by SETDB1 and both SETDB1 and SUV39H1/2 in mESCs, respectively. To determine whether this trend is specific to pluripotent stem cells, we analyzed H3K9me2 profiles in iMEFs. As there were no publicly available Hi-C data for iMEFs, we also performed Hi-C analysis for WT control iMEFs. Similar to mESCs, H3K9me2 was more enriched in B than in A compartments in iMEFs (Supplementary Fig. 2a). Compared with mESCs, the larger size of the H3K9me2 domains was preserved by UNC0642 treatment: the average size of H3K9me2 domains was 54.5 kb in iMEFs and 14.2 kb in mESCs (Fig. 2a, b). Interestingly, a marked reduction of H3K9me2 in the A compartments was observed in UNC0642-treated iMEFs, but not in the B compartments. H3K9me2 decreased upon UNC0642 treatment in 91.1% of A compartments, but only decreased in 23.9% of B compartments (Fig. 2a, c, Supplementary Fig. 2b). G9a/GLP-independent H3K9me2 domains in iMEFs occupied 88.5% of the B compartments and 31.0% of the A compartments (Fig. 2d). Therefore, it can be said that H3K9me2 in the B compartments was more resistant to UNC0642 treatment than that in the A compartments in iMEFs.

We also performed H3K9me2 ChIP-seq analysis on UNC0642-treated *Setdb1* KO iMEFs, which have been established previously[25]. Further reduction of H3K9me2 in the A compartments was observed in UNC0642-treated *Setdb1* KO iMEFs (Supplementary Fig. 2c). We found that 79.1% of the A compartments and 45.2% of the B compartments had lower H3K9me2 levels in UNC0642-treated *Setdb1* KO iMEFs than in UNC0642-treated WT iMEFs (Fig. 2e, Supplementary Fig. 2d).

The number and total length of the H3K9me2 domains were decreased in the A compartments of UNC0642-treated *Setdb1* KO iMEFs, whereas those in the B compartments remained largely unchanged (Fig. 2f). Therefore, SETDB1 is essential for G9a/GLP-independent H3K9me2 in the A compartments in iMEFs, as observed in mESCs.

To determine whether SETDB1 and SUV39H1/2 mediate all G9a/GLP-independent H3K9me2 in iMEFs, we established *Setdb1* and *Suv39h1/2* TKO iMEFs using the CRISPR-Cas9 system (Supplementary Fig. 2e). These TKO iMEFs had undetectable levels of H3K9me3 by western blotting (Supplementary Fig. 2f). H3K9me2 was undetectable after UNC0642 treatment in *Setdb1* and *Suv39h1/2* TKO iMEFs (Supplementary Fig. 2g). The complete loss of H3K9me2 in UNC0642-treated TKO iMEFs was also confirmed using mass spectrometry (Supplementary Fig. 2h). ChIP-qPCR of the selected regions in the B compartments, shown in Supplementary Fig. 1o, also showed a loss of H3K9me2 in UNC0642-treated TKO iMEFs (Fig. 2g). Thus, we concluded that H3K9me2 in the B compartments is redundantly regulated by G9a/GLP, SETDB1, and SUV39H1/2, in both mESCs and iMEFs.

**G9a/GLP-independent H3K9me2 is correlated with efficient H3K9me2 recovery.** Heterochromatin can spread along chromatin from nucleation sites, such as recruiters of the writer molecule binding regions[26,27]. As G9a and GLP bind to nucleosomes that contain H3K9me2 via ankyrin repeats of G9a and GLP[28] and methylate adjacent nucleosomes[29], we hypothesized that H3K9me2 spreads from G9a/GLP-independent H3K9me2 sites during H3K9me2 domain formation. To test this hypothesis, we analyzed H3K9me2 recovery in mESCs after UNC0642 removal (Fig. 3a). As shown in Supplementary Fig. 3a, H3K9me2 levels nearly recovered three days after UNC0642 was withdrawn. Therefore, we performed H3K9me2 ChIP-seq at each time point after UNC0642 withdrawal (0, 24, 32, 40, 48, 56, 64, and 72 h). From H3K9me2 ChIP-seq time-course data, we obtained H3K9me2 dynamics during the recovery period (Fig. 3b). Principal component analysis of H3K9me2 reads per kilobase million (RPKM) in the 80-kb genomic window showed that the H3K9me2 profile gradually approached that of WT after UNC0642 withdrawal, although had not completely recovered 72 h after withdrawal (Fig. 3c). To investigate the role of G9a/GLP-independent H3K9me2 in this recovery, we classified genomic regions as "early," "middle," or "late," based on the timing of H3K9me2 recovery after UNC0642 withdrawal, comparing

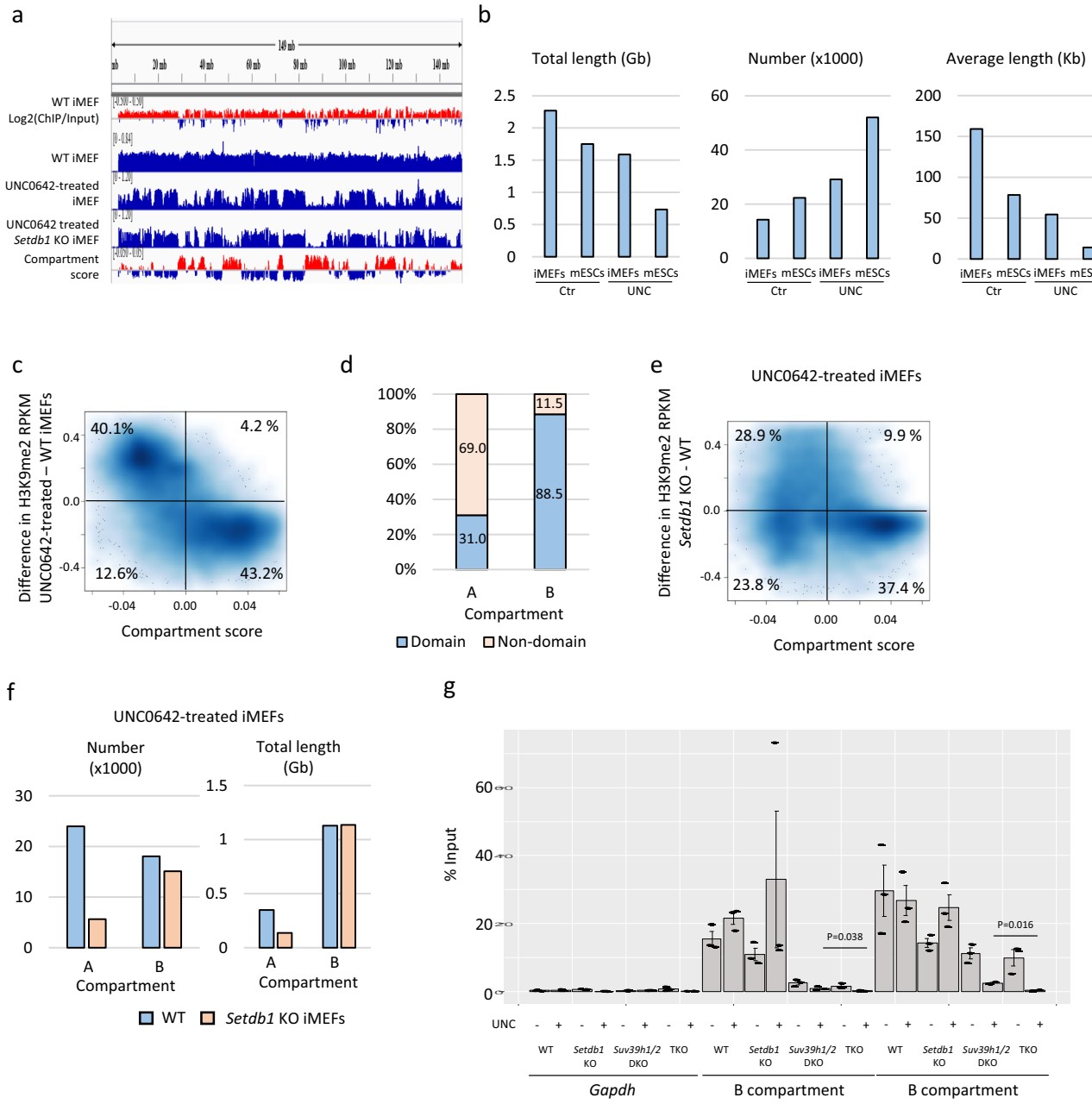

**Fig. 2 Characterization of SETDB1- and SUV39H1/2-dependent H3K9me2 regions in iMEFs. a** Representative view of H3K9me2 ChIP-seq data along chromosome 6 in iMEFs. Although H3K9me2 in the A compartments is decreased by UNC0642 treatment, low H3K9me2 signal is retained. Those retained H3K9me2 is further diminished in UNC0642-treated *Setdb1* KO iMEFs. **b** Difference in G9a/GLP-independent H3K9me2 between mESCs and iMEFs. H3K9me2 domain size is larger in iMEFs than in mESCs. **c** A comparison between compartment score and H3K9me2 changes by UNC0642 treatment in iMEFs. Overall, H3K9me2 reduction in the A compartments is induced by the UNC0642 treatment. Each plot represents data from each 80-kb bin. Darker blue represents higher dot density. **d** Fractions of H3K9me2 domains in the A and B compartments in UNC0642-treated iMEFs. **e** A comparison between compartment score and H3K9me2 changes in UNC0642-treated *Setdb1* KO iMEFs. In comparison with UNC0642-treated WT iMEFs (**c**), UNC0642-treated *Setdb1* KO iMEFs showed further reduction of H3K9me2, especially in the A compartments. **f** The number and the length of H3K9me2 domains in WT and *Setdb1* KO iMEFs treated with UNC0642. Both the number and the length of H3K9me2 domains in the A compartments are reduced in UNC0642-treated *Setdb1* KO iMEFs. **g** H3K9me2 ChIP-qPCR analysis in regions showing in Supplementary Fig. 1o. H3K9me2 in the region 1 and 2 is completely lost in UNC0642-treated TKO iMEFs. Data are mean ± SEM; *n* = 3.

H3K9me2 levels among genomic region classes in UNC0642-treated mESCs (see Materials and Methods). In this analysis, regions with similar H3K9me2 levels in WT mESCs were compared. The "early" class showed a higher H3K9me2 level after UNC0642 treatment (Fig. 3d, Supplementary Fig. 3b) and overlapped more frequently with G9a/GLP-independent H3K9me2

and SETDB1-dependent H3K9me2 than the "middle" and "late" classes (Fig. 3e, f). In addition, H3K9me2 recovery after UNC0642 removal was not induced well in *Setdb1* KO mESCs and TKO iMEFs (Supplementary Fig. 3c, d). To investigate whether H3K9me2 can spread from G9a/GLP-independent H3K9me2 during recovery, we analyzed H3K9me2 profiles

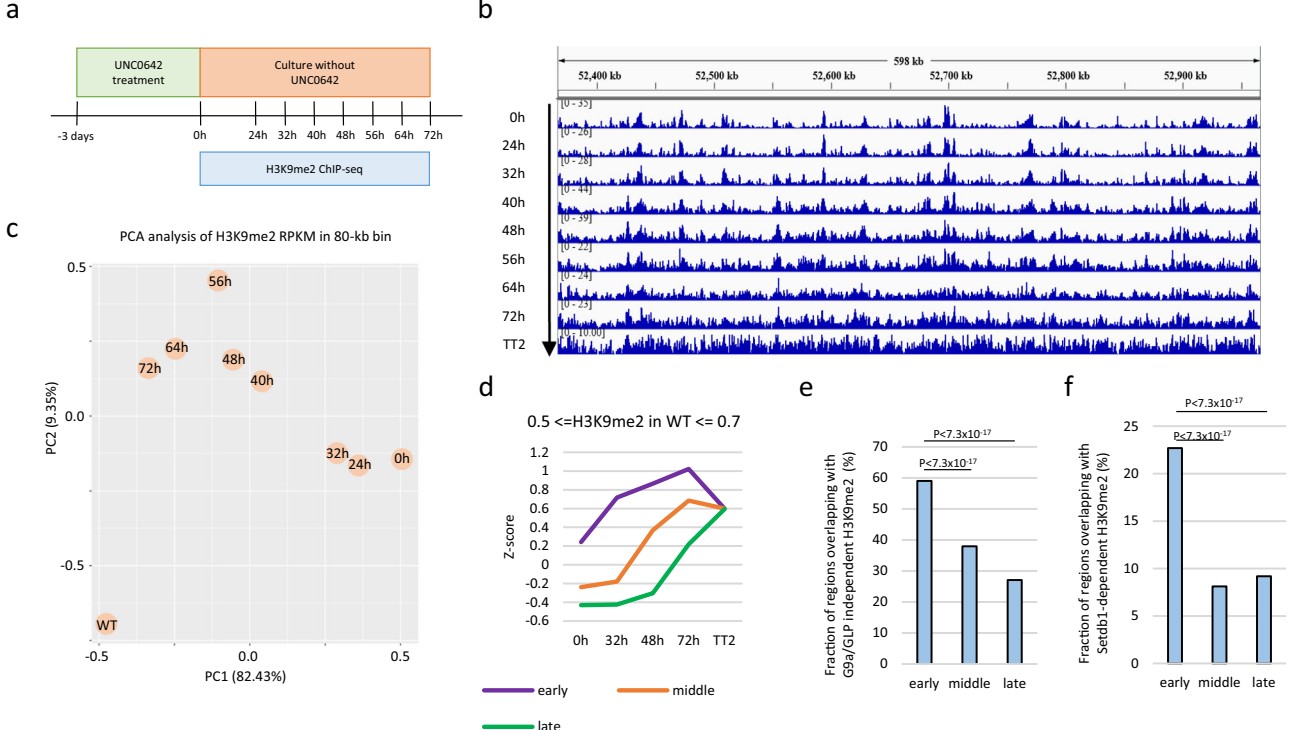

**Fig. 3 Recovery of H3K9me2 after the UNC0642 treatment. a** An experimental design to investigate H3K9me2 recovery. mESCs were treated with UNC0642 for 3 days, and then H3K9me2 ChIP-seq was performed at each time point after the UNC0642 removal. **b** A representative view of H3K9me2 ChIP-seq data during H3K9me2 recovery. **c** A principal component analysis of H3K9me2 ChIP-seq data during H3K9me2 recovery. **d** Distinct kinetics of H3K9me2 recovery from UNC0642 treatment between three classes of sequences. Each 10-kb region throughout the genome was classified into three types named "early", "middle", and "late" based on the timing of H3K9me2 recovery. Z-scaling was performed on H3K9me2 RPKM of 10-kb bins at each time point. $Y$ axis represents the average $Z$ scores of each class. Only regions with $Z$ scores between 0.5 and 0.7 in WT mESCs were used for this analysis to match the final H3K9me2 levels among three classes. **e, f** Overlap between G9a/GLP-independent H3K9me2 domains (**e**) or SETDB1-dependent H3K9me2 domains (**f**) and each class. Significant enrichment of G9a/GLP-independent or SETDB1-dependent H3K9me2 in early class was calculated by hypothesis testing for the difference in the population proportions using a function of prop.test by R.

around G9a/GLP-independent H3K9me2 domains during recovery. However, we were not able to obtain crucial evidence of H3K9me2 spreads from H3K9me2 domains (Supplementary Fig. 3e). Therefore, G9a/GLP-independent H3K9me2 might help in fast recovery of H3K9me2, but H3K9me2 spreads from G9/GLP-independent H3K9me2 are limited.

**G9a/GLP-independent H3K9me2 is involved in transcriptional repression**. Next, we determined whether G9a/GLP-independent H3K9me2 regulates transcription. From RNA-seq of WT and *Setdb1* KO mESCs treated with or without UNC0642, we identified 495, 194, and 79 upregulated genes and 61, 3, and 0 downregulated genes in UNC0642-treated *Setdb1* KO, *Setdb1* KO, and UNC0642-treated mESCs, respectively (Supplementary Fig. 4a). Only 3.5, 1.4, and 0.5% of genes that harbor G9a/GLP-independent H3K9me2 within 5 kb of TSSs were upregulated in these three mESCs, suggesting that G9a/GLP-independent H3K9me2 were mostly dispensable for transcriptional silencing. There were 267 of 495 upregulated genes in UNC0642-treated *Setdb1* KO mESCs that were neither identified in *Setdb1* KO without UNC0642 treatment nor in UNC0642-treated WT mESCs, suggesting redundancy of SETDB1 and G9a/GLP in gene repression (Supplementary Fig. 4a). G9a/GLP primarily repressed genes in the B compartments, whereas SETDB1 repressed genes in both A and B compartments (Supplementary Fig. 4b). Only 27 genes were commonly upregulated in UNC0642-treated mESCs and *Setdb1* KO mESCs (Supplementary Fig. 4a). However, among

the upregulated genes in UNC0642-treated mESCs (2 + 50 + 27 = 79 in Supplementary Fig. 4a), 72 genes (16 and 56 in the A and B compartments, respectively) were derepressed in *Setdb1* KO mESCs at least twice (Supplementary Fig. 4c), suggesting that most genes repressed by G9a/GLP-dependent H3K9me2 are also regulated by SETDB1. In contrast, among upregulated genes in *Setdb1* KO mESCs, those in the B compartments were more derepressed in UNC0642-treated mESCs than genes in the A compartments (35 of 103 in the A compartments (26.9%) and 51 of 90 in the B compartments (56.6%), fold change ≥2) (Supplementary Fig. 4d). Furthermore, the additive upregulation observed following UNC0642 treatment in *Setdb1* KO mESCs was higher in genes in the B than in the A compartments (Supplementary Fig. 4e). Therefore, G9a/GLP-mediated H3K9me2 plays a more critical role in silencing of genes repressed by SETDB1 in the B than in the A compartments in mESCs.

To investigate the association between G9a/GLP-independent H3K9me2 and gene expression, we performed an integrative analysis of RNA-seq, H3K9me2 ChIP-seq, and H3K9me3 ChIP-seq data from WT and *Setdb1* KO mESCs treated or not treated with UNC0642. Both G9a/GLP-independent H3K9me2 and H3K9me3 were frequently found within 5 kb from the transcriptional start sites (TSSs) of upregulated genes in each condition (Fig. 4a). In addition, G9a/GLP-independent H3K9me2 covered both up- and downstream of TSSs of upregulated genes in *Setdb1* KO mESCs, whereas H3K9me3 covered only upstream of TSSs (Fig. 4b, c), suggesting that G9a/GLP-independent H3K9me2 is

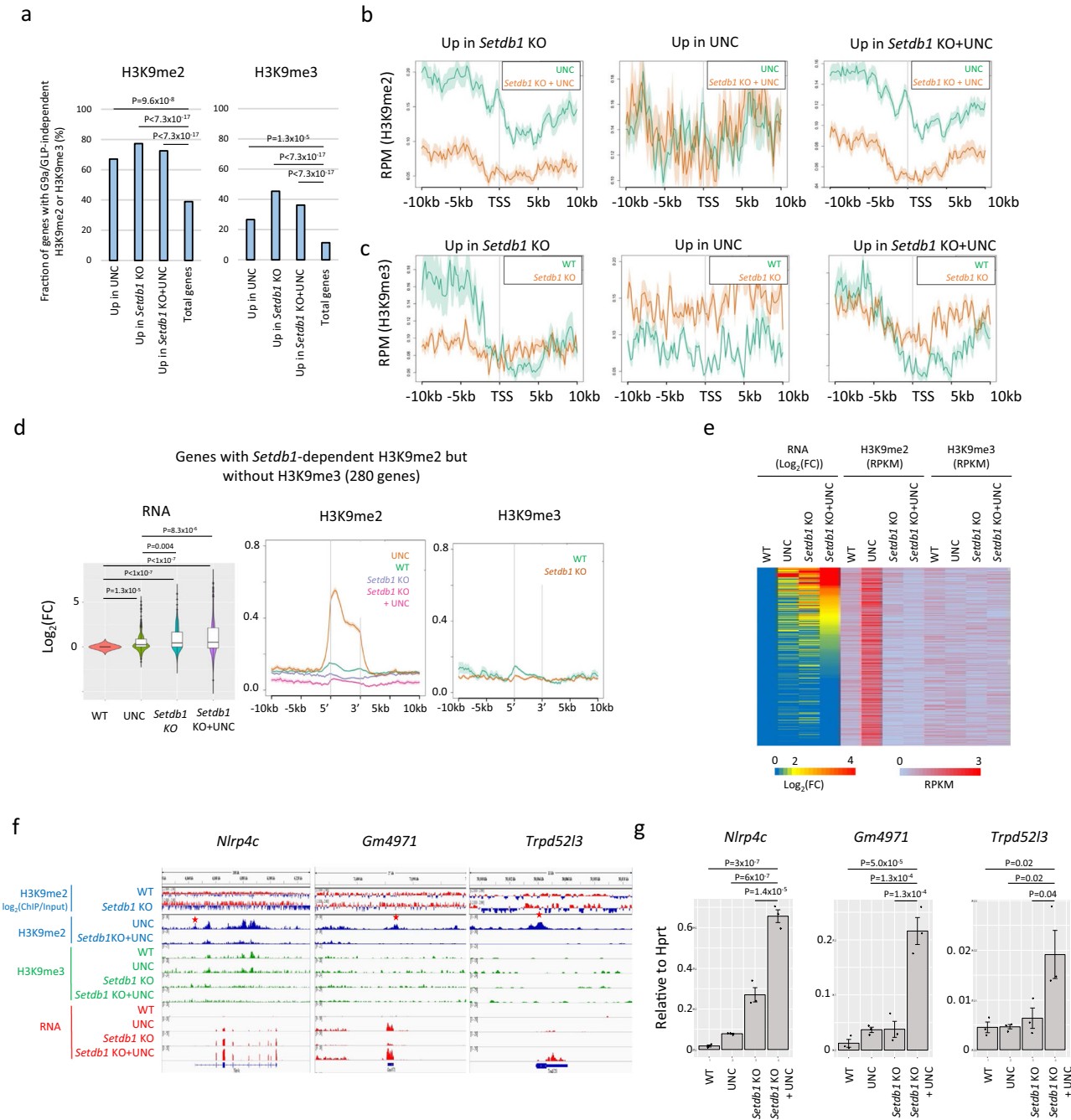

**Fig. 4 Function of G9a/GLP-independent H3K9me2 in transcriptional regulation. a** Overlap of upregulated genes with G9a/GLP-independent H3K9me2 or H3K9me3 domains. The graph shows the fractions of upregulated genes in each condition that overlap with G9a/GLP-independent H3K9me2 domains or H3K9me3 domains within 5 kb of TSS. Significant enrichment of H3K9me2 around TSS was calculated by hypothesis testing for the difference in the population proportions using a function of prop.test in R. **b**, **c** Enrichment of H3K9me2 in WT or *Setdb1* mESCs treated with UNC0642 (**b**) and enrichment of H3K9me3 in WT or *Setdb1* KO mESCs (**c**) around upregulated genes in each condition. G9a/GLP-independent H3K9me2 is enriched from upstream of TSS to gene body of upregulated genes identified in *Setdb1* KO mESCs treated with or without UNC0642. This H3K9me2 is largely dependent on SETDB1. **d** Expression and H3K9 methylation profiles of genes with SETDB1-dependent H3K9me2 but without H3K9me3. The boxplot shows log2 fold change of expression. The Middle and right figures show the enrichment of H3K9me2 and H3K9me3 around the TSS in WT or *Setdb1* KO mESCs treated with or without UNC0642, respectively. **e** Heatmap of fold change of expression, H3K9me2 RPKM, and H3K9me3 RPKM in genes with H3K9me2 but without H3K9me3. H3K9me2 and H3K9me3 RPKM were calculated ~0.5 kb of the TSS. **f** Representative genes with G9a/GLP-independent H3K9me2 but without H3K9me3. H3K9me2, H3K9me3, and RNA-seq profiles of Nlrp4c, Gm4971, and Trpd52l3. H3K9me2 in WT and *Setdb1* KO mESCs shows as log2(ChIP/Input). Star mark represents SETDB1-dependent H3K9me2 around TSS. **g** qRT-PCR of representative genes shown in Fig. 4f. Bar graph shows the relative expression of these three genes in each condition ($N = 3$). $P$ value was calculated by Dunnett's test.

not just a transitional stage leading to H3K9me3. To determine whether SETDB1-dependent H3K9me2 functions as a transcriptional repressor, the expression of 280 genes marked with SETDB1-dependent H3K9me2 but no H3K9me3 around TSS was analyzed. H3K9me2 in *G9a/GLP* DKO mESCs was also enriched around these genes, which suggests that enrichment of H3K9me2 was not an artifact of the inhibitor (Supplementary Fig. 4f). Furthermore, as shown in Fig. 4d, the H3K9me2 on these genes was lower in *Setdb1* KO mESCs than in control cells. Thus, G9a/GLP-independent, SETDB1-dependent H3K9me2 is actually present in WT cells. Although these genes were not clearly marked with H3K9me3, they were upregulated in *Setdb1* KO mESCs both with or without UNC0642 (Fig. 4d, e). The upregulation of the three selected genes (*Nlrp4c*, *Gm4971*, and *Trpd52l3*) (Fig. 4f) in UNC0642-treated *Setcb1* KO mESCs was validated using qRT-PCR (Fig. 4g). Additive increased expression caused by UNC0642 treatment in *Setdb1* KO was observed for all three genes, consistent with the redundancy of G9a/GLP and SETDB1-mediated H3K9me2 in transcriptional silencing. These data support the idea that SETDB1-dependent H3K9me2 also has a role in transcriptional repression.

**Correlation of decreased H3K9me2 with reorganization of the active compartment setting**. To clarify whether H3K9me2 has a role in 3D genome organization, we performed Hi-C analysis for each H3K9 MTase-deficient mESC line, both with and without UNC0642 treatment, calculating compartment scores in bins of 100 kb. The Hi-C analysis showed an overall maintenance of compartment patterns in all samples (Fig. 5a), with more than 94% of compartment profiles conserved in all samples (Fig. 5b). Moreover, overall compartment scores did not differ significantly between samples (Supplementary Fig. 5a). To elucidate the effect of the H3K9me2 changes on the nuclear compartments more specifically, we compared the changes in H3K9me2 in UNC0642-treated mESCs and UNC0642-treated *Setdb1* KO mESCs to those of the compartment scores. The genomic regions with compartment scores increased by UNC0642 had decreased H3K9me2 levels (Fig. 5c) and decreased fractions of H3K9me2 domains (Fig. 5d). Consistent with this, genomic regions with decreased fractions of H3K9me2 domains had increased compartment scores (Fig. 5e). The regions with "B-to-A" conversion showed a decreased fraction of H3K9me2 domains (Fig. 5f) and those with decreased fractions of H3K9me2 domains frequently showed "B-to-A" conversions (Fig. 5g). The correlation between the changes in compartment score and H3K9me2 above was reproducible (Supplementary Fig. 5b–f). To validate whether the correlation between changes in compartment score and H3K9me2 actually occurred in *G9a/GLP* DKO mESCs, we reanalyzed recently published Hi-C and H3K9me2 ChIP-seq data from *G9a/GLP* DKO mESCs[30]. This correlation between increased compartment score and decreased H3K9me2 was also observed in *G9a/GLP* DKO mESCs (Supplementary Fig. 5b–f). A representative region showing the correlation between decreased H3K9me2 and increased compartment score is shown in Supplementary Fig. 5g. Finally, we investigated the correlation between gene activation and 3D genome organization. The genes upregulated in UNC0642-treated WT or *Setdb1* KO mESCs showed a significant increase in compartment scores (Fig. 5h). Therefore, decreased H3K9me2 correlated with the relocation of the target genes to more active compartments; moreover, this relocation is correlated with transcriptional activation, although the overall compartment pattern is maintained in H3K9me2-depleted cells (Fig. 5i).

Overall, this study demonstrated that H3K9me2 is regulated by different sets of H3K9 MTases between the A and B compartments and that G9a/GLP-independent H3K9me2 plays a role in

efficient H3K9me2 recovery, transcriptional repression, and 3D genome organization (Fig. 6).

## Discussion

This study investigated the role of five different H3K9 MTases in H3K9me2 domain formation and the function of G9a/GLP-independent H3K9me2 in transcriptional repression and 3D genome organization. Although many studies have implicated the presence of G9a/GLP-independent H3K9me2 in mESCs and iMEFs[6,14,15], its mechanism and function are unknown. G9a/GLP-independent H3K9me2 has been reported to be enriched in pericentromeric satellite repeats in mESCs[6,14]. It also has been observed in both H3K9me3-marked regions in mESCs and late-replicating domains in MEFs[15]. Therefore, it has been predicted that G9a/GLP-independent H3K9me2 is mediated by other H3K9 MTases, such as SETDB1 and SUV39H1/2.

We formally demonstrated that G9a/GLP-independent H3K9me2 is mediated by SETDB1 and SUV39H1/2. Surprisingly, the contribution of SETDB1 and SUV39H1/2 to H3K9me2 differed between nuclear compartments. SETDB1 was essential for G9a/GLP-independent H3K9me2 in the A compartments, whereas both SETDB1 and SUV39H1/2 mediate this process in the B compartments (Fig. 6a); however, it remains unknown why SUV39H1/2 function is restricted only to the B compartments. A mixture of heterochromatin protein 1 (HP1)-SUV39H1-TRIM28 complexes, H3K9me2, and three-marked nucleosomal arrays undergoes phase separation in vitro[31]. HP1 and SUV39H1 are enriched in DAPI-dense regions and form droplet-like structures in cultured cells[32]. Thus, one possible explanation for the SUV39H1 restriction to the B compartments is phase separation; another possible reason is the genomic distribution of SUV39H1/2-target regions. SUV39H1/2 represses LINE1 retrotransposon, and LINE1 is enriched in AT-rich isochores and gene-poor regions, consistent with the features of the B compartment[33]. Thus, the enrichment of LINE1 in B compartments might restrict SUV39H1/2 function in this location.

Heterochromatin has been shown to spread from nucleation sites. G9a and GLP can bind to H3K9me1 and H3K9me2 via their ankyrin repeat domain[28], and GLP might pre-methylate nucleosomes and further methylate the premethylated neighboring nucleosomes[29]. We demonstrated that H3K9me2 reduction following UNC0642 treatment recovered more efficiently after inhibitor removal in regions with high G9a/GLP-independent H3K9me2 (Fig. 6b). However, we could not obtain crucial evidence of H3K9me2 spreads from G9a/GLP-independent H3K9me2 (Supplementary Fig. 3e). Thus, G9a/GLP-independent H3K9me2 facilitates efficient H3K9me2 recovery, but its spreading might be limited. The efficient recovery of genomic regions marked with G9a/GLP-independent H3K9me2 might be supported by ANK-domain-mediated pre-bound G9a/GLP to H3K9-methylated nucleosomes upon UNC0642 removal. We found that SETDB1 depletion resulted in no H3K9me2 recovery in mESCs (Supplementary Fig. 3c). The HP1-CAF1-SETDB1 complex monomethylates K9 on non-nucleosomal histone H3, and H3K9me1 is imposed during translation by SETDB1[34]. Thus, H3K9me1 on non-nucleosomal histone H3 mediated by SETDB1 may also provide a scaffold for G9a/GLP during H3K9me2 domain establishment. As H3K9me2 domains are established in intergenic regions in mouse oocytes by G9a, and the global H3K9me2 level is decreased in *Setdb1* KO oocytes[35], SETDB1-mediated G9a/GLP-independent H3K9me2 and/or SETDB1-mediated non-nucleosomal H3K9me1 may be important for H3K9me2 domain establishment during gametogenesis.

In addition to H3K9me2 recovery, our transcriptome analysis demonstrated that G9a/GLP-independent H3K9me2 has a role in transcriptional repression. Some SETDB1-target genes harbor SETDB1-dependent H3K9me2 but not H3K9me3 (Fig. 4d–f). It is

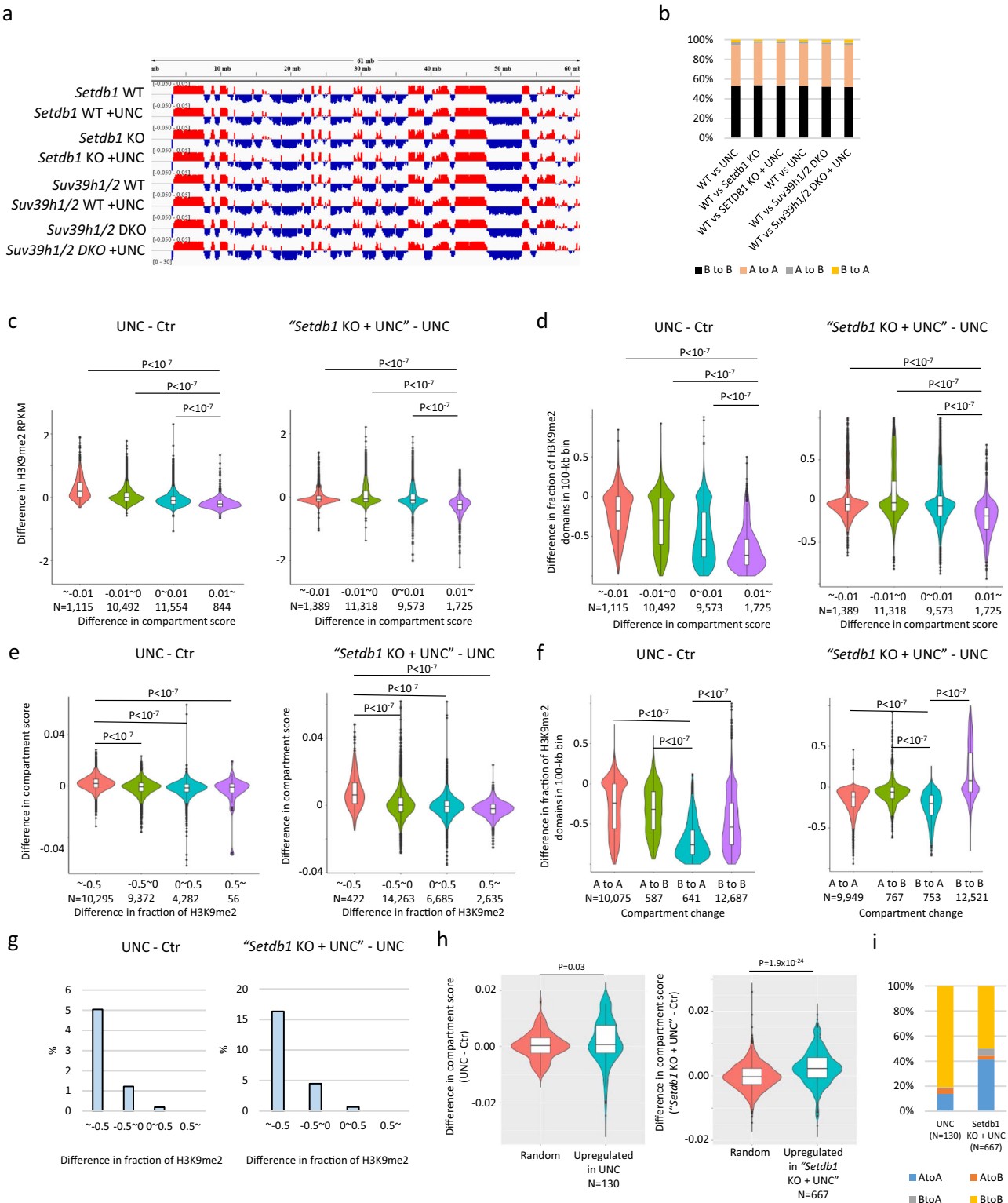

unknown why histone methylation patterns differ among SETDB1-target genes. As SETDB1 catalytic activity is regulated by ATF7IP and the monoubiquitylation of lysine 867 of the hSETDB1, monoubiquitylation of SETDB1 or ATF7IP enrichment at the target genomic regions might alter the catalytic activity of SETDB1[4,36]. Another possible mechanism to generate G9a/GLP-independent H3K9me2 is demethylation of H3K9me3 by H3K9 demethylase such as KDM4A[37], and enrichment of SETDB1-dependent H3K9me2 without H3K9me3 downstream of

TTSs of the SETDB1-target genes (Fig. 4b, c) might be caused by demethylation of H3K9me3.

Our Hi-C analysis showed that overall nuclear compartment patterns were maintained in all analyzed samples. However, decreased H3K9me2 upon UNC0642 treatment in mESCs correlated well with increased compartment scores (Fig. 5c–f). Therefore, we propose that H3K9 dimethylation contributes to the inactive compartment setting in mammals (Fig. 6c). Because UNC0642-treated *Setdb1* KO mESCs still possess H3K9me2 in

**Fig. 5 Correlation of decreased H3K9me2 and movement toward more active compartment. a** Representative view of compartment score in each 100-kb bin in mESCs. The regions colored by red and blue represent the A and B compartments, respectively. **b** Compartment change in each condition. More than 94% of compartments are maintained in all conditions. **c** Violin plots showing differences in H3K9me2 RPKM in each 100-kb bin classified by a degree of compartment score changes in UNC0642-treated mESCs. $P$ value was calculated by Tukey's test. Statistics comparison was only shown between the bins with $\geq 0.01$ (0.01~) difference in compartment score and others. **d** Violin plots showing differences in H3K9me2 domain fraction in each 100-kb bin classified by a degree of compartment score changes in UNC0642-treated mESCs. $P$ value was calculated by Tukey's test. Statistics comparison was only shown between the bins with $\geq 0.01$ (0.01~) difference in compartment score and others. **e** Violin plots showing changes of compartment scores in each 100-kb bin classified by a degree of changes in H3K9me2 domain fraction in UNC0642-treated mESCs. $P$ value was calculated by Tukey's test. Statistics comparison was only shown between the bins with $<-0.5$ (~$-0.5$) difference in fraction of H3K9me2 and others. **f** Violin plots showing changes of fraction of H3K9me2 domain in 100-kb bin classified by change of compartment pattern in UNC0642-treated mESCs. $P$ value was calculated by Tukey's test. Statistics comparison was only shown between the bins classified "B to A" and others. **g** Bar graph showing the fraction of B to A compartment changes in each 100-kb bin classified by the difference in H3K9me2 domain fraction in UNC0642-treated mESCs. **h** Violin plot showing changes in compartment score around upregulated genes in UNC0642-treated mESCs and randomly selected genes. Increased compartment score is observed in upregulated genes in UNC0642-treated mESCs. $P$ value was calculated by one-tailed Student's $t$ test. **i** Pattern of compartment changes in upregulated genes in UNC0642-treated mESCs.

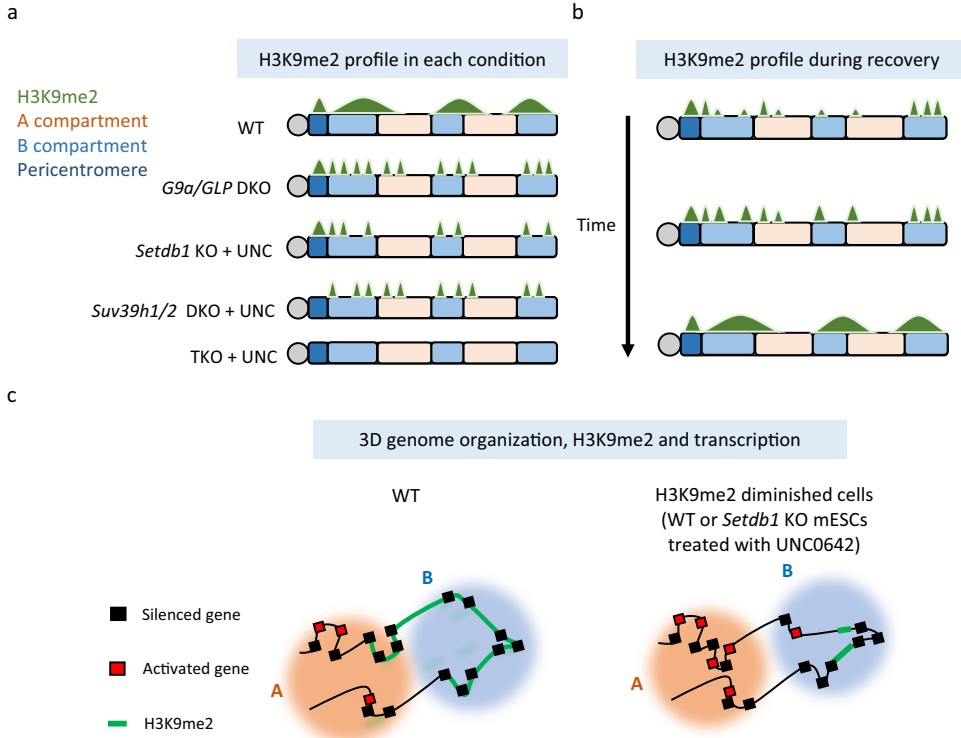

**Fig. 6 Summary of this study. a** Although H3K9me2 (shown in green) is largely mediated by G9a/GLP in mESCs, some residual H3K9me2 is observed in *G9a/GLP* DKO mESCs. SETDB1 and SUV39H1/2 mediate G9a/GLP-independent H3K9me2 in a compartment-dependent manner. SETDB1 and SUV39H1/2 are essential for G9a/GLP-independent H3K9me2 in the A compartments (orange) and in pericentromeric satellite repeats (dark blue), respectively, whereas H3K9me2 in the B compartments (light blue) is mediated by both SETDB1 and SUV39H1/2 redundantly. **b** During H3K9me2 domain establishment after recovery from a UNC0642 treatment, H3K9me2 is recovered efficiently in regions with high G9a/GLP-independent H3K9me2. Thus, G9a/GLP-independent H3K9me2 might facilitate the efficient establishment of H3K9me2 domains. **c** We found that the reduction of H3K9me2 (WT or *Setdb1* KO treated with UNC0642) resulted in a movement toward a more active compartment setting, which was sometimes accompanied by transcriptional activation. Thus, we propose that G9a/GLP-independent H3K9me2 functions in H3K9me2 domain establishment, transcriptional repression, and 3D genome organization.

the B compartments (Fig. 6a), but H3K9me2 and H3K9me3 are almost depleted in UNC0642-treated *Setdb1, Suv39h1/2* TKO mESCs and iMEFs, it will be of interest to analyze genome compartmentalization in this cell line in order to fully dissect the role of H3K9 methylation in this process.

## Methods

**Cell culture**. We used previously established *Setdb1* cKO mESCs[8], *G9a/GLP* DKO mESCs[7], *G9a* KO mESCs[6], *GLP* KO mESCs[7], *Suv39h1/2* WT and DKO mESCs[10], *Setdb1* KO iMEFs[25], and *Suv39h1/2* DKO iMEFs[32]. mESCs were maintained in Dulbecco's modified Eagle's medium (DMEM, Sigma, D6429) containing 10% Knockout

SR (Invitrogen, 10828-028), 1% fetal bovine serum (Biowest, S1820), MEM non-essential amino acids solution (Gibco, 11140-050), 0.1% LIF and 2-mercaptoethanol (Nacalai tesque, 21417-52) described as ES medium hereafter). Mouse embryonic fibroblasts were maintained in DMEM (Nacalai tesque, 08458-16) containing 10% fetal bovine serum (Biosera, FB1061), MEM non-essential medium, and 2-mercaptoethanol (Nacalai tesque, 21417-52). To inhibit G9a/GLP catalytic activity, mESCs and iMEFs were cultured for 3 days with 2 μM UNC0642. For KO of *Setdb1* in mESCs, *Setdb1* cKO mESCs were cultured for 5 days with 800 nM 4-OHT, whereas *Setdb1* cKO/*Suv39h1/2* DKO mESCs were cultured for 7 days with 800 nM 4-OHT.

**Establishment of *Setdb1, Suv39h1/2* TKO cells**. *Setdb1* cKO/*Suv39h1/2* DKO mESCs: *Setdb1* cKO mESC, #33-6[8] was transfected with the following four gRNAs:

1. *Suv39h1* exon 3 upstream in PL-CRISPR.EFS.tRFP (Addgene, #57819), 2. *Suv39h2* exon 4 downstream in pKLV2-U6.gRNA (Bbs1)-PGK.puro-BFP[38], 3. *Suv39h2* exon 3 in PL-CRISPR.EFS.tRFP, 4. *Suv39h2* exon 4 in pKLV2-U6.gRNA (Bbs1)-PGK.puro-BFP. Three days after the transfection, BFP and RFP double-positive cells were sorted by flow cytometry (BD FACSAria). From the sorted cells, *Suv39h1/2* DKO mES clone #4 was identified using PCR and western blotting. Partial deletion of SUV39H1/2 SET domains was validated by RNA-seq analysis (Supplementary Fig. 1k). For making *Setdb1*, *Suv39h1/2* TKO mESCs, *Setdb1* cKO/ *Suv39h1/2* DKO mESCs were treated with 800 mM 4-OHT for 7 days as described.

*Setdb1*, *Suv39h1/2* TKO iMEFs: *Setdb1* KO iMEFs [25] were transfected with *Suv39h1* exon 4 gRNA in pKLV2-U6.gRNA(Bbs1)-PGK.puro-BFP and *Suv39h2* exon 4 upstream and downstream gRNAs in pX330-BB[39] and selected with puromycin. From the puromycin-resistant cells, *Setdb1*, *Suv39h1* DKO iMEF clone #27 was identified by Western blot analysis. Then, *Suv39h1* KO clone #27 was further transfected with *Suv39h2* exon 3 gRNA in pL-CRISPR.EFS.tRFP and pKLV2-U6.gRNA(Bbs1)-PGK.puro-BFP and selected with Puromycin again. From the puromycin-resistant cells, *Setdb1*, *Suv39h1/2* TKO KO cell clone #27-34 was identified by Western blot analysis. CRISPR-Cas9 mediated *Suv39h1* and *2* gene mutations were confirmed by DNA sequencing analysis (Supplementary Fig. 2e).

**Native ChIP**. At least $2 \times 10^5$ cells were lysed in 50 µl Buffer 1 (60 mM KCl, 15 mM NaCl, 5 mM MgCl$_2$, 0.1 mM EGTA, 15 mM Tris-HCl (pH7.5), 0.3 M Sucrose, 0.5 mM DTT, and protease inhibitors (Nacalai tesque, 04080-11), then were added 50 µl Buffer2 (Buffer 1 + 1% NP40) to the sample. After the incubation for 10 min on ice, the sample was added 800 µl Buffer 3 (60 mM KCl, 15 mM NaCl, 5 mM MgCl$_2$, 0.1 mM EGTA, 15 mM Tris-HCl (pH7.5), 1.2 M Sucrose, 0.5 mM DTT, and protease inhibitors). After centrifugation at 9000 rpm for 10 min at 4°C, the sample was added 100 µl MNase buffer (0.32 M Sucrose, 50 mM Tris-HCl (pH7.5), 4 mM MgCl$_2$, 1 mM CaCl$_2$, and phenylmethylsulfonyl fluoride (PMSF)) followed by incubation for 20 min at 37 °C with 0.3 U MNase (TAKARA, 2910 A). After the incubation, the reaction was stopped by 10 µl 0.5 M EDTA. After centrifugation at 15,000 rpm for 10 min at 4°C, the supernatant was added 900 µl Incubation buffer (50 mM NaCl, 20 mM Tris-HCl (pH7.5), 5 mM EDTA, 0.01% NP40, ad PMSF). In all, 10% of the sample was used for input, and the remains proceeded the following procedure. The sample was mixed with the antibody-beads complex which was formed by incubation of 20 µl protein A/ G beads (Santa Cruz, sc-2003) with antibody in Incubation buffer on ice for 1 h. After rotating for overnight at 4°C, the complex was washed by 500 µl Wash buffer A (75 mM NaCl, 50 mM Tris-HCl (pH7.5), 10 mM EDTA, and 0.01% NP40), 500 µl Wash buffer B (100 mM NaCl, 50 mM Tris-HCl (pH7.5), 10 mM EDTA, and 0.01% NP40), and 500 µl wash buffer C (175 mM, 50 mM Tris-HCl (pH7.5), 10 mM EDTA, and 0.01% NP40). The ChIP DNA was eluted by RNase and Protease treatment followed by DNA purification using PCR purification kit (QIAGEN).

**Western blotting**. In brief, cells were suspended in radioimmunoprecipitation buffer (50 mM Tris-HCl (pH 8.0), 420 mM NaCl, 0.5% sodium deoxycholate, 0.1% Sodium dodecyl sulfate, 1% NP40) and sonicated. The extract was incubated for 30 min on ice, and then incubated at 95 °C for 5 min. The extract was loaded and run on sodium dodecyl-sulfate polyacrylamide gel electrophoresis (SDS-PAGE) gel as standard protocols. For histone proteins, intensity was analyzed by OdysseyR CLx Imagins System (LI-COR).

**Mass spectrometry analysis for H3K9 methylation**. Histones were prepared by the acid-extraction method as previously described[6]. The samples were subjected to SDS-PAGE and stained with Coomassie blue. The protein bands corresponding to histone H3 were excised from the gel and were digested with *Achromobacter* protease I (Lys-C) in gel. The digests were performed nano-liquid chromatography— tandem mass spectrometry using EASY-nLC 1200 liquid chromatography (Thermo Fisher Scientific) connected to Q-Exactive HFX mass spectrometer equipped with nanospray ion source (Thermo Fisher Scientific). Peptides containing the digests were separated with a linear gradient of 0-100%B buffer in A (A:0.1% formic acid, B:80%acetonitrile/0.1%formic acid) for 20 min with a reversed-phase column (NTCC analytical column, C18, φ75 µm × 150 mm, 3 µm; Nikkyo Technos, Japan). MS and MS/MS data were acquired using a data-dependent TOP 10 method. Protein quantification was performed with Proteome Discoverer 2.4. (Thermo Fisher Scientific) using a sequence database search node as a MASCOT program 2.7 (Matrix Science) with following parameters, Database: Histone 130611 (66 sequences; 15002 residues), Enzyme: Lys-C/P, Variable modifications: Acetyl (Protein N-term),Oxidation (M),Gln->pyro-Glu (N-term Q),Acetyl (K),Methyl (K),Trimethyl (K),Dimethyl (K), Mass values: Monoisotopic, Protein Mass: Unrestricted, Peptide Mass Tolerance: ±15 ppm, Fragment Mass Tolerance: ±30 mmu, Max Missed Cleavages: 3, Instrument type: ESI-TRAP. Protein methylation rates were obtained peak areas of selected ion chromatograms that are the tetra charged protonated molecules at m/z 381.9706 (methyl), m/z 385.4745 (dimethyl), and m/z 388.9784 (trimethyl) of "QTARKSTGGKAPRK"-related peptides (1xAcetyl [K10]; 1xGln->pyro-Glu [N-Term]; 1xmethylations [K5]) using Qual Browser (Thermo xcalibur 4.1.50, Thermo Fisher Scientific) with ±15-ppm width.

**RT-qPCR**. RNA was isolated by RNeasy Plus Mini Kit (Qiagen) following manufacturer's instructions. cDNA synthesis was performed with Omniscript RT Kit

(Qiagen). qPCR was carried out using Power SYBR Green PCR Master Mix (ThermoFisher Scientific) on StepOnePlus$^{TM}$ (ThermoFisher Scientific). The signals were normalized relative to *Hprt*. All qPCR data are represented as the mean ±standard deviation of three biological replicates.

**Preparation of ChIP-seq library**. The ChIP DNA was fragmented by Picoruptor (Diagenode) for 10 cycles of 30 s on, 30 s off. Then, ChIP library was constructed by KAPA Hyper Prep Kit (KAPA BIOSYSTEMS) and SeqCap Adapter Kit A (Roche) or SMARTER ThruPLEX DNA-seq kit (TAKARA) and SMARTer DNA Unique Dual Index Kit (TAKARA) according to manufacturer instructions. The concentration of the ChIP-seq library was quantified by KAPA Library quantification kit (KAPA BIOSYSTEMS). ChIP-sequencing was performed on a HiSeq X platform (Illumina). We performed two biological replicates for H3K9me2 ChIP-seq and correlation between replicates as described in Supplementary Table 1.

**Preparation of RNA-seq library**. In all, 500 ng of total RNA was used for RNA-seq library construction. RNA-seq library was constructed by KAPA mRNA Hyper Prep Kit (KAPA BIOSYSTEMS) and SeqCap Adapter Kit (Roche) according to manufacturer's instructions. The concentration of the RNA-seq library was quantified by KAPA Library quantification kit (KAPA BIOSYSTEMS). mRNA sequencing was performed on a HiSeq X platform (Illumina). We performed two biological replicate for RNA-seq and correlation between replicates as described in Supplementary Table 1.

**Preparation of HiC-seq library**. Hi-C experiments were performed as previously described[40,41], based on DpnII enzyme (4-bps cutter) using $2 \times 10^6$ fixed cells. Hi-C libraries were subject to paired-end sequencing (150 base pair (bp) read length) using HiSeq X Ten. Detailed protocol for HiC-seq library preparation is available at Protocols.io (https://www.protocols.io/view/iconhi-c-protocol-ver-1-0-4mjgu4n). We performed two biological replicates for Hi-C seq and correlation between replicates as described in Supplementary Table 1.

**Hi-C data processing and A/B compartment calculation**. Hi-C data processing was done by using Docker for 4DN Hi-C pipeline (v43, https://github.com/4dn-dcic/ docker-4dn-hic). The pipeline includes alignment (using the mouse genome, mm10) and filtering steps. After filtering valid Hi-C alignments,.hic format Hi-C matrix files were generated by Juicer Tools[42] using the reads with MAPQ > 10. The A/B compartment (compartment score) profiles (in 100-kb bins) in each chromosome (without sex chromosome) were calculated from.hic format Hi-C matrix files (intrachromosomal KR normalized Hi-C maps) by Juicer Tools[42] as previously described[43].

**Mapping and domain identification of ChIP-seq data**. Adaptor sequences in reads were trimmed using Trim Galore version 0.3.7 (http://www.bioinformatics. babraham.ac.uk/projects/trim_galore/). Then trimmed reads were aligned to the mm10 genome build using bowtie version 0.12.7[44] with default parameters. Duplicated reads were removed using samtools version 0.1.18[45]. Enriched or depleted status of H3K9me2 in each 2 kb was determined by *Hiddendomains* and continuous H3K9me2 enriched 2 kb regions was connected as H3K9me2 domain with some modifications: the option of Bin size and max.read.count was 2000 bp and 150, respectively. Read number in each bin was normalized by the following formula to adjust difference in read number among samples. Normalized read number = read number × 30,000,000/total read number. For the hidden Markov parameter to determine enriched and depleted states, the average parameter among chromosomes rather than the parameter calculated by each chromosome was used, because very few domains were identified in some chromosomes. Loss of H3K9me2 domains was identified using getDifferentialPeaks in Homer (fold change ≥ 4 and P value ≤ 0.0001). Conserved H3K9me2 domains ware the domains where was identified also in a sample and were not identified as lost domains.

**Classification of ChIP-seq regions by the timing of H3K9me2 recovered**. RPKM of each 10-kb bin was calculated and the RPKM was converted to z score. The 10-kb bin where Z score increased by 0.3 or more at 32 h, 48 h, and 72 h from 0 h was annotated as early, middle, and late, respectively.

**Identification of genes with *Setdb1*-dependent H3K9me2 but without H3K9me3 from ChIP-seq data**. Genes that satisfy the following criteria are extracted: 1. H3K9me2 domains are located within 0.5 kb of TSS in UNC0642-treated mESCs, but not in UNC0642-treated *Setdb1* KO mESCs. 2. H3K9me3 domains are not located within 0.5 kb of TSS in any condition. 3. H3K9me2 RPKM in UNC0642-treated mESCs ~0.5 kb of TSS is >1, and that in UNC0642-treated *Setdb1* KO mESCs is <1. 4. H3K9me3 RPKM ~0.5 kb of TSS is less than one in any condition.

**Handling ChIP-seq biological replicate data**. We performed two biological replicates for H3K9me2 ChIP-seq except for H3K9me2 recovery. The correlation of biological replicates was described in Supplementary Table 1. Because correlation between replicates was high and compartment-dependent regulation of H3K9me2 by *Setdb1* and *Suv39h1/2* was reproducible, H3K9me2 domain analysis was performed on replicate 1 data. For visualizing ChIP-seq data, we also used replicate 1 data.

**RNA-seq analysis**. Raw FastQ data were trimmed with Trim Galore (v0.3.7, default parameters) (http://www.bioinformatics.babraham.ac.uk/projects/trim_galore/) and mapped to the mouse GRCm38 genome assembly using TopHat (v2.1.1)[46]. After read mapping, mapped reads were analyzed by TEtranscripts (v1.4.11, default parameters)[47]. We performed two biological replicates for RNA-seq and identified DE genes by DESeq2 (adj. $P$ value < 0.05, FC > 10)[48]. Average expression levels of replicates were used for the downstream analysis.

**Visualization of NGS data**. The Integrative Genomics Viewer was used to visualize NGS data. Enrichment of H3K9me2/H3K9me3 enrichment in specified regions was visualized by ngsplot. Scatter plot analysis, principal component analysis boxplot and violin plot analysis were performed by R script.

**Immunofluorescence analysis**. In all, $2 \times 10^4$ cells were seeded on laminin-coated 12-well Chamber (Ibidi) the day before fixation. The cells were fixed with 4% paraformaldehyde for 10 min at room temperature, permeabilized with 0.1% Triton X-100 for 10 min, blocked with 3% bovine serum albumin 0.1% Tween20 in phosphate-buffered saline and incubated overnight with primary antibodies at 4 °C. Anti-mouse IgG conjugated with Alexa Fluor 568 (ThermoFisher Scientific) or anti-rabbit IgG conjugated with Alexa Fluor 488 (ThermoFisher Scientific) were used as secondary antibodies. The nuclei were counterstained with DAPI, observed under fluorescence microscopy and analyzed Olympus FluoView™ FV3000 (Olympus).

**Antibodies**. For western blotting, antibodies specific for histone H3 (07-690, EMD Millipore), H3K9me3 (2F3)[49], H3K9me2 (6D11[50] and 39754, Active Motif), histone H3 for Supplementary Fig. 2g (96C10, Cell Signaling), SETDB1 (Cp10377, Cell Applications), SUV39H1 (#8729, CST), SUV39H2 (LS-116360, LSBio), G9a (A8620A)[7] and GLP (B0422B)[7] were used as primary antibody. For histone proteins, IRDye 800CW Goat anti-Mouse IgG (926-32210, LI-COR) and IRDye 680RD Goat anti-Rabbit IgG (926-68071, LI-COR) were used as secondary antibody. For non-histone proteins, HRP-linked anti-Rabbit IgG (NA934, GE Healthcare) and HRP-linked anti-Mouse IgG (NA931, GE Healthcare) were used as secondary antibody. For immunofluorescence analysis, H3K9me2 (6D11), H3K9me3 (39161, Active Motif) were used for primary antibody, and Goat anti-Mouse IgG Alexa Fluor 568 (A-11031, Invitrogen) and Goat anti-Rabbit IgG Alexa Fluor 488 (A-11034) were used for secondary antibody. For ChIP analysis, antibodies specific for H3K9me3 (2F3) and H3K9me2 (6D11) were used. Specificity of antibodies for H3K9me2 and H3K9me3 were previously reported[49,50].

**Oligonucleotides**. Oligonucleotide sequences for PCR primers and production of gRNA targeting vectors are listed in Supplementary Tables 2 and 3, respectively.

**Statistics and reproducibility**. The data presented as mean ± SEM were tested for significance in the paired $t$ test. Post hoc tests were conducted using the Dunnett or Tukey. All error bars represent SEM were calculated as $\frac{std(x)}{\sqrt{n}}$ for numeric variables.

All western blotting and qPCR were performed for three times. IF experiments were repeated twice. H3K9me2 ChIP-seq, RNA-seq, and Hi-C were repeated twice, except for H3K9me2 recovery experiment.

**Reporting summary**. Further information on research design is available in the Nature Research Reporting Summary linked to this article.

## Data availability
Source data for main figures have been provided in Supplementary Data 1. All reads from the RNA-seq, ChIP-seq, and Hi-C experiments generated in this study have been submitted to Gene Expression Omunibus under accession number GSE169106. Read number of NGS data are listed in Supplementary Table 4. The mass spectrometry proteomics data have been deposited to the ProteomeXchange Consortium with the dataset identifier PXD024756. H3K9me2 ChIP-seq and Hi-C data from *G9a/Glp* DKO mESCs are available from GEO under accession number GSE138102[30].

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

## Acknowledgements

We thank Thomas Jenuwein (Max Planck Institute) for providing *Suv39h1/2* DKO mES and iMEF cell lines, Tomohiro Yamazaki and Tetsuo Hirose for providing dual gRNA expression vector PX330-B/B, the staff of the Support Unit for Bio-Material Analysis (BMA) at RIKEN Center for Brain Science (CBS) Research Resources Division (RRD) for DNA sequencing, and NGS library construction with special thanks to K. Ohtawa. We also thank all of the Cellular Memory laboratory members for helpful discussions. This research was supported by KAKENHI (18H03991 and 18H05530) and a RIKEN internal research fund to Y.S., and KAKENHI (19K16049) and the Special Postdoctoral Researcher (SPDR) Program of RIKEN to K.F.

## Author contributions

Conceptualization: Y.S., K.F. Cell culture and western blotting analysis: C.S., K.F. NGS analysis: K.F., A.T., H.M., H.I. Mass spectrometry analysis: T.S., N.D.. Paper writing: Y.S., K.F., I.H., T.D.

## Competing interests

The authors declare no competing interests.
