## [Peer Review File · Communications Biology]

Reviewers' comments:

Reviewer #1 (Remarks to the Author):

This manuscript by Fukuda and colleagues addresses the contribution of H3K9 methyltransferases to H3K9me2 generation specifically, as well as the role of H3K9me2 in transcriptional regulation and compartment formation. These questions are pertinent and the insight generated by this manuscript is potentially important to the fields of chromatin biology and nuclear organization. The contribution of the different methyltransferases to H3K9me2 in ES cells is largely robustly concluded. However the contribution of H3K9me2 to transcriptional regulation provided is insufficiently demonstrated in my view (see my comments to Figure 4). The contribution of H3K9me2 to compartment formation is the most important conclusion in this manuscript in my view. However the data in Figure 5 to support this is limited. This part of the manuscript should be extended (see my specific comments to Figure 5).

My major criticism of this manuscript is that the reproducibility of the data is not reported. N numbers are missing throughout the manuscript. Please report the number of times each experiment was repeated or from how many independent samples the data was generated, in the Figure legends. Quantifications and statistical testing should also be included where appropriate – please see my individual comments.

Major comments

Figure 1D. The length of H3K9me2 domains in Setdb1 knockouts decreased slightly in the A compartment. Here the authors conclude that this suggests Setdb1 has a role in K9me2 domain formation. However the number of domains is not reported in the two knockout conditions. If the number of domains are similar but the size decreases, I would argue this suggests Setdb1 has a role in domain expansion/spreading rather than formation. Please report the number of H3K9me2 domains in both knockout conditions in A and B compartments.

Figure 1E. Why is the control of UNC-treated ES cells not included here? While the authors suggest that the profile of G9a/GLP DKO ES cells resembles that of UNC-treated cells in just one snapshot, the correlation of these two samples should be presented, with statistical testing. If they are indeed not non-identical, the proper control for Setdb1 KO + UNC treatment should be used, which is UNC treatment alone, to determine the effect of Setdb1 here.

Figure 2A/B. A decrease in K9me2 domain size in response to UNC treatment in iMEFs as the authors conclude is not clear, as the size of domains is not reported in untreated iMEFs. Figure 2A suggests a decrease in H3K9me2 domain size in UNC treated iMEFs for the region shown. Figure 2B only shows UNC treated iMEFs. Please provide complete domain parameters (size, number) for untreated and UNC-treated iMEFs and ES cells to determine whether the decrease in domain size in ES cells also occurs in iMEFs.

Figure 2C. The compartment scores from ES cells were used for this analysis in iMEFs, which are likely to be unequal to those in iMEFs. Thus the compartment annotations for iMEFs may not be correct, which obscures the conclusion as to whether H3K9me2 changes in A or B compartments. Please use compartment scores for the iMEF cell line used in this study. If the data is not available the authors should perform Hi-C to determine compartment locations in the iMEFs, otherwise this data regarding compartment effects in iMEFs should be removed from the manuscript in my view.

Fig. S2E. The quantification does not have any error bars, suggesting that it comes from just one experiment (The n numbers are not reported in the legend). Please repeat the experiment 3 times and provide quantification with error bars and statistical testing.

Fig. S3A, C-D. The quantifications do not have any error bars, suggesting that they come from just one experiment (The n numbers are not reported in the legends). Please repeat the experiments 3 times and provide proper quantification with error bars and statistical testing.

Figure 3C suggests that the recovered H3K9me2 at 72h does not match with WT cells (considering the 9% variance explained by PC2), suggesting that either not all H3K9me2 recovers in this time frame or the recovery occurs in different locations. The conclusion that 'As H3K9me2 almost

completely recovered three days after 174 UNC0642 removal' should be modified accordingly.

Figure 4. Given the widespread occurrence of H3K9me2 across the genome (Fig. 1C) the reported numbers of upregulated genes are minimal, suggesting a limited effect of H3K9me2 on transcriptional regulation. The number of downregulated genes should also be reported. While the data shown in Figure 4A suggest that upregulated genes are often proximal to H3K9me2 domains this is not surprising given the prevalence of H3K9me2, even in UNC-treated cells (Fig. S1C). The authors should provide controls for this analysis. Are the upregulated genes more likely to be within 5KB to G9a/GLP-independent H3K9me2 domains than non-changed genes or downregulated genes? What % of G9a/GLP independent K9me2 domain genes (within 5KB of the TSS) are upregulated/unchanged/downregulated?

The authors' statement that H3K9me2 correlated with gene repression more than H3K9me3 is too general and misleading. A more correct conclusion would be that H3K9me2 correlated more with gene repression of UNC-dependent genes than H3K9me3. This is of course not surprising and exhibits circular reasoning. These genes were already shown to be regulated by H3K9me2 (upregulated upon UNC treatment), thus the fact that they have stronger enrichment of H3K9me2 than me3 is expected. Any comparison between the repressive effects of H3K9me2 and me3 cannot be conducted unless a proper analysis is performed removing K9me2 from K9me2-only regions and K9me3 from K9me3-only regions.

Figure 4D. No statistics are presented for the expression plot, thus it is not possible to determine which group is significantly upregulated. In Figure 4E it appears that these genes do not possess H3K9me2 in control cells and only gain H3K9me2 in UNC treated cells. The H3K9me2 levels in control cells should also be plotted in Figure 4D. This also means that expression differences should be compared between Setdb1 KO + UNC to UNC-treated controls in both Figure 4D and F, rather than to . It is not clear to which specific groups the authors refer when they suggest they observe upregulation, and most importantly, without statistics again it is impossible to conclude.

Furthermore, this suggests that the acquisition of H3K9me2 on these genes is an artifact of the inhibitor and thus not physiologically relevant. Can the authors provide evidence that G9a/GLP independent, Setdb1-dependent H3K9me2 actually occurs in wild-type cells? If not then analysis of the transcriptional effects of the loss of this modification must be interpreted with caution. Do these regions also acquire H3K9me2 in G9a/GLP knockouts for example?

Figure 5. The data demonstrate a clear correlation between H3K9me2 loss in UNC-treated cells and an increase in compartment score. As this is an important conclusion this analysis should be extended. First in order to validate this observation, Hi-C data should be generated from G9a/GLP DKO ES cells to determine if a similar trend is observed. Secondly, the authors mention that increases in compartment scores are observed in both A and B compartments, but the data is not presented. Please add this analysis to the Figure. Thirdly, they should correlate compartment scores to H3K9me2 domains. In UNC treated cells domains are smaller. How does this correlate to compartment scores? Do compartments also shrink in size? Do strongly affected H3K9me2 domains result in compartment switching or more pronounced differences in compartment scores? Do the few switching compartment lose H3K9me2? Finally the authors should present Hi-C plots of a representative chromosome/genomic region along with associated H3K9me2 profiles to show compartment weakening, correlating to H3K9me2 loss.

Minor comments

Reference 39: Invalid citation

Fig. S1A Ponso should be Ponceau

Figure 1b. An increase rather than a decrease in the H3K9me2 signal is observed in compartment A in Setdb1KO and in compartment B in Suv39h1/2 DKO. Is the y axis labelled incorrectly? Also applies to Figure 1F.

Fig. S1C please provide a quantification of H3K9me2 levels from 3 independent experiments

Fig. S1D please report the number of times the experiment was repeated.

Fig. S1H The legend is missing.

Fig. S1L and M please provide quantification of H3K9me2 and 3.

Reviewer #2 (Remarks to the Author):

The paper by Fukuda et al. examines the role of H3K9me2 and H3K9me3 in gene regulation and chromatin conformation in mouse ESCs and iMEFs. While multiple previous studies have addressed some of these questions the authors try to unravel the relative contribution of five H3K9 methyltransferases in putting down H3K9me2 and silencing genes. By using multiple genetic mutants in combination with a specific G9a inhibitor the authors claim that there is significant amount of G9a/GLP-independent H3K9me2. This is mainly dependent on Setdb1 and to a lesser extent Suv39h1/2. This G9a/GLP-independent H3K9me2 correlates with gene repression and regulates chromatin compartmentalization. These claims are novel and likely of interest for the readers of *Communication Biology*. Indeed, by performing meticulous chromatin based assays in a panel of mutants the authors uncover the importance of G9a/GLP-independent H3K9me2. However in my opinion there are some important weaknesses to this manuscript and the main conclusions are to some extent misleading.

Nevertheless I feel that this is an elegant and thorough study, after addressing my comments I support the publication of this manuscript. Below I'm attaching my detailed criticism.

Major points:

-In general the experiments are well designed. However I feel like the data analysis is sometimes misleading. It is clear from the correlations that in the vast majority of G9a-independent H3K9me2 does not regulate gene expression or chromatin conformation. This mark is very wide-spread and yet very few genes become derepressed. This was also reported for G9a-dependent H3K9me2 (Zylicz et al, 2015). Please provide quantitative data on what fraction of G9a-independent H3K9me2 targets are actually regulated by this mark? The authors never show how many genes are downregulated in all the mutants, this data has to be shown. The authors use constitutive KOs which will accumulate secondary gene expression changes, this should at least be discussed. The authors should provide a clear conclusion acknowledging that for most targets H3K9me2 does not seem to be functional.

-RNAseq analysis throughout the paper constantly uses different thresholds for Fold Change (e.g. Figure s4b; Figure s4c; and in page 10 line 222). Why is this? Data analysis should be unified and key conclusions should be validated using multiple thresholds. It is vital to have at least duplicates of RNAseq experiments to perform robust analysis.

-Figure 4: in this analysis the authors claim that H3K9me2 correlates better with gene repression. However the analysis is missing key controls. How will the analysis look at random genes throughout the genome? H3K9me2 is much more prevalent than H3K9me3, are the correlations observed actually significant when compared to remaining parts of the genome?

-The authors claim that H3K9me2 regulates chromatin compartmentalisation however the observed changes are relatively weak. Can they be fully explained by transcriptional upregulation? Are compartmentalisation scores affected also on H3K9me2 targets that do not get upregulated in mutants? If not, then it seems unlikely that H3K9me2 actually regulates chromatin compartments, it is rather affected by gene expression. To strengthen this point the authors should do HiC in TKO cells treated with UNC and see if the conformation is more affected. Conclusion on p14 l.298-299 is not supported by the data and has to be rewritten. There is no convincing data that H3K9me2 is one of the key players of determinant mechanism.

-The authors do not actually analyse H3K9me2 spreading upon UNC removal. They convincingly uncover that regions with more G9a-independent H3K9me2 recover faster but do not look at spreading from these regions. This analysis should be performed, how do windows directly bordering G9a-independent domains behave? Also the use of UNC, a catalytic inhibitor, implies that G9a/GLP can still play a function in quick re-establishing of H3K9me2. Indeed, it is possible

that G9a/GLP is inactive but pre-bound to regions which recover faster upon UNC removal. This possibility should be discussed. Conclusion on spreading (e.g. p13 I279) needs to be amended.

- Conclusion from p6 I126-7 claims a big contribution of Suv39h1/2 in mediating H3K9me2 however the ChIPseq data did not reveal this (Fig 1H). As authors suggest this is because the repetitive fraction of the genome could be mainly affected. As the data stands the conclusions are not supported by the analysis. Authors should include H3K9me2 ChIPseq analysis of the repetitive fraction of the genome. Alternatively the conclusions should be significantly softened.

-Statistical analysis is missing throughout the manuscript especially in multiple analysis of genome-wide datasets.

-The title is not informative and does not provide any insight into the conclusions of this paper.

Minor comments:

-Model in Figure 6 proposes that Setdb1KO+Unc treatment results in relocation of genes away from nuclear lamina- this has not been addressed in this paper.

-Figure S4C- what are the diagonal lines representing?

-Immunoblot staining in e.g. Fig 1 is not spelled Ponso but Ponceau S

-H3K9me2 genome wide pattern is better visualised when windows are shown as $\text{Log}_2(\text{H3K9me2}/\text{Input})$, this was done in many publications e.g. Lienert et al, 2011.

-Nora et al, 2012 should be cited along Dixon et al for TADs.

Reviewer #3 (Remarks to the Author):

In "Functional correlation of H3K9me2 and nuclear compartment formation" Fukuda and colleagues map H3K9me2 in mESC and MEF cells genome wide to determine which methyltransferases regulate deposition of H3K9me2 in different genomic compartments. The authors determine that Setdb1 (thought to be primarily an H3K9me3 methyltransferase) is required for methylation of H3K9me2 in the A compartment of the genome. The authors performed Hi-C to map changes in the compartments cause by loss and/or inhibition of the 5 methyltransferases and find that largely the compartments remain unchanged. The authors' work attempts to address a fundamental question in chromatin biology of how histone modifications regulate the establishment and maintenance of 3D genome architecture, an important question in the field. However, as currently constructed the manuscript fall short of providing evidence for this concept.

The manuscript title "Functional correlation of H3K9me2 and nuclear compartment formation" Is not supported by the authors own data in figure 5, where ~95% of chromatin Hi-C domains do not change when the authors manipulate the H3K9me2 methyltransferases. The authors do not demonstrate formation of any compartments is dependent on H3K9me2 or any of the methyltransferases. While the observations of which methyltransferases are responsible for methylation at different genomic regions is important, this does not satisfy the authors title and goals.

While the authors do provide convincing evidence that Setdb1 is required for some H3K9me2 deposition, the authors do not provide evidence that the enzyme is directly depositing H3K9me2. The authors do not consider the role of the demethylases in the formation of the H3K9me2 domains. The experiments in Suv39H DKO and SETDB1 KO cells yielding reduced H3K9me2 domains could be because there is reduced H3K9me3 there is less target for demethylation to H3K9me2. Rather than the methyltransferases producing H3K9me2 directly. The authors should, at a minimum, moderate their language throughout to reflect this. However, addressing if the demethylases are important would significantly strengthen the manuscript.

The authors need to provide validation (either their own or published for the monoclonal H3K9me2 and H3K9me3 antibodies used for ChIP) that they really are specific for dimethyl or trimethyl modifications as this has historically been a big problem on the cross reactivity of these antibodies.

Minor Points:

Many figures need to have improved labeling throughout. Many graphs are lack statistical analysis. The methods need to be improved with more details for the sequencing approaches. In particular, how many replicates were performed and how were the replicates combined to produce the tracks presented. Many of the sequencing tracks need to have higher resolution images as well.

The sentence on lines 79-81 is very confusing and I had to read it 3 times to understand what the authors meant when compared to the data. My initial reaction reading the sentence and looking at the figure was that the authors were wrong, but after carefully rereading 5 times I understood the intent. Please clarify the writing.

Please include error bars on all bar graphs of western blot quantification and ensure that all blots have quantification.

Figure 1D- please present this as a box and whisker or violin plot to clearly indicate the median and quartiles. Is this statistically different than wild type cells?

Figure 1E- the authors should analyze the cells by flow cytometry to ensure the cells are in equal distributions of the cell cycle. If the UNC treated cells arrest (or the G9aGLP DKO) cells in a different phase this could explain the clustering and domain differences.

Figure 1H, please include in the figure a label along the X axis to indicate what the heatmaps are in relation to.

Figure 2B- Please use box and whisker or violin plots to plot the distribution to provide more information than the basic bar plots.

Figure 3C- "Principal component analysis of H3K9me2 Reads Per Kilobase Million (RPKM) in the 80-kb genomic window showed gradual H3K9me2 recovery after UNC0642 removal (Fig. 3C)" This is not true, this simply says the timepoints have different principal component values. It says nothing about recovery of H3K9me2. If the authors want to make this point, they should plot a metagene analysis centered on the H3K9me2 peaks at time point 0 and plot average H3K9me2 levels up and downstream from this point (maybe +/-100 KB? Or region at authors discretion) at each of the subsequent time points.

Line 237- "correlation was mild" No correlation coefficient by any correlation measure was provided in figure 5. To make this statement please provide a proper correlation analysis. Was the analysis in Figure 5C-D only performed on compartments with changes (i.e only 5% of compartments?) Figure 5 seems vastly overinterpreted, the simplest conclusion is that H3K9me2 did not affect compartment formation.

Fig S1C- please provide quantification of the amount of reduction. It's not clear what Pan-H3 is for an antibody, is this acetylated H3 or just recognizing all H3 in which case just label it H3. If it is supposed to be the loading control, then the UNC treated samples are clearly reduced in histones and another loading control should be chosen.

Figure S1D- please include a scale bar on all micrograph images.

Figure S1F is blurry and a low-resolution screenshot from IGV. Please also clearly indicate the signal and the domain tracks as was done in the main figures.

Figure S2D spell Ponceau correctly. Why does H3K9me3 increase in the Setdb1 ko cells?

Figure S4- what do the lines emanating from the corner of the violin plots mean?

Reviewers' comments:

Reviewer #1 (Remarks to the Author):

This manuscript by Fukuda and colleagues addresses the contribution of H3K9 methyltransferases to H3K9me2 generation specifically, as well as the role of H3K9me2 in transcriptional regulation and compartment formation. These questions are pertinent and the insight generated by this manuscript is potentially important to the fields of chromatin biology and nuclear organization. The contribution of the different methyltransferases to H3K9me2 in ES cells is largely robustly concluded. However the contribution of H3K9me2 to transcriptional regulation provided is insufficiently demonstrated in my view (see my comments to Figure 4). The contribution of H3K9me2 to compartment formation is the most important conclusion in this manuscript in my view. However the data in Figure 5 to support this is limited. This part of the manuscript should be extended (see my specific comments to Figure 5).

My major criticism of this manuscript is that the reproducibility of the data is not reported. N numbers are missing throughout the manuscript. Please report the number of times each experiment was repeated or from how many independent samples the data was generated, in the Figure legends. Quantifications and statistical testing should also be included where appropriate – please see my individual comments.

Major comments

Figure 1D. The length of H3K9me2 domains in *Setdb1* knockouts decreased slightly in the A compartment. Here the authors conclude that this suggests *Setdb1* has a role in K9me2 domain formation. However the number of domains is not reported in the two knockout conditions. If the number of domains are similar but the size decreases, I would argue this suggests *Setdb1* has a role in domain expansion/spreading rather than formation. Please report the number of H3K9me2 domains in both knockout conditions in A and B compartments.

Response: Thank you for a valuable comment. Indeed, We found increased number of the H3K9me2 domains in the A compartments of *Setdb1* KO mESCs. Therefore, we added this information and changed the sentence in page 5, line 85-89 as bellow. “The total length of H3K9me2 domains in the A compartments slightly decreased in *Setdb1* KO mESCs (Fig. 1D), while the number of such domains was slightly increased (10,879 in WT and 12,022 in *Setdb1* KO), suggesting that SETDB1 plays a role in H3K9me2 domain expansion/spreading in A compartments.”

Figure 1E. Why is the control of UNC-treated ES cells not included here? While the authors suggest that the profile of G9a/GLP DKO ES cells resembles that of UNC-treated cells in just one snapshot, the correlation of these two samples should be presented, with statistical testing. If they are indeed not non-identical, the proper control for *Setdb1* KO + UNC treatment should be used, which is UNC treatment alone, to determine the effect of *Setdb1* here.

Response: *Suv39h1/2* DKO and its control WT mESC lines were established from blastocyst outgrowth (Lehnertz et al., 2003) and *Setdb1* cKO mESC line (#33-6) we used in this study was established by gene targeting in mESC line TT2 (Matsui et al., 2010). Therefore, we included two cell lines as WT control for *Setdb1* KO and *Suv39h1/2* DKO mESCs, namely *Setdb1* WT (*Setdb1* cKO before OHT treatment) and *Suv39h1/2* WT, respectively. H3K9me2 profiles of “*Setdb1* WT + UNC” and “*Suv39h1/2* WT + UNC” were the control for “*Setdb1* KO + UNC” and “*Suv39h1/2* DKO + UNC”, respectively. These control data were included in Fig. 1E and these were well correlated with H3K9me2 profile of *G9a/GLP* DKO mESCs (Fig. 1E). We also showed the correlation of H3K9me2 profiles between UNC0642-treated *Setdb1* WT and *G9a/GLP* DKO mESCs as bellow (only for the reviewer).

Figure 2A/B. A decrease in K9me2 domain size in response to UNC treatment in iMEFs as the authors conclude is not clear, as the size of domains is not reported in untreated iMEFs. Figure 2A suggests a decrease in H3K9me2 domain size in UNC treated iMEFs for the region shown. Figure 2B only shows UNC treated iMEFs. Please provide complete domain parameters (size, number) for untreated and UNC-treated iMEFs and ES cells to determine whether the decrease in domain size in ES cells also occurs in iMEFs.

Response: The reviewer’s criticism is absolutely correct. We added the number, size and average size of H3K9me2 domains in UNC-untreated iMEFs and ESCs to the revised Fig. 2B, thus UNC-treatment decreased total length and average length of the H3K9me2 domains and increased total number of the domains in both mESCs and iMEFs.

Figure 2C. The compartment scores from ES cells were used for this analysis in iMEFs, which are likely to be unequal to those in iMEFs. Thus the compartment annotations for iMEFs may not be correct, which obscures the conclusion as to whether H3K9me2 changes in A or B compartments. Please use compartment scores for the iMEF cell line used in this study. If the data is not available the authors should perform Hi-C to determine compartment locations in the iMEFs, otherwise this data regarding compartment effects in iMEFs should be removed from the manuscript in my view.

Response: Thank you. This criticism is also absolutely correct. We performed Hi-C analysis for WT iMEFs and replaced Fig. 2C-F and Fig. S2A-D with data using iMEF’s

compartment score. We also changed some numerical values related to those figures in the manuscript. However, our conclusion was not changed.

1. "Similar to mESCs, H3K9me2 was more enriched in B than in A compartments in iMEFs (Additional file: Fig. S2A)." page 8, line 140-141.

2. "Further reduction of H3K9me2 in A compartments was observed in UNC0642-treated *Setdb1* KO iMEFs (Additional file 1: Fig. S2C)." page 8, line 152-153.

3." a marked reduction of H3K9me2 in A compartments was observed in UNC0642-treated iMEFs, but not in B compartments." page 8, line 144-145.

4. original statement "It was observed that 80.8% of the A compartments (32.8/7.8 + 32.8) and 48.7% of the B compartments (28.9/30.5 + 28.9) showed lower H3K9me2 levels in the UNC0642-treated *Setdb1* KO iMEFs than in the UNC0642-treated WT iMEFs (Fig. 2E)."

revised statement " We found that 79.1% of the A compartments and 45.2% of the B compartments had lower H3K9me2 levels in UNC0642-treated *Setdb1* KO iMEFs than in UNC0642-treated WT iMEFs (Fig. 2E, Additional file 1: Fig. S2D)." page 8, line 153-155.

Fig. S2E. The quantification does not have any error bars, suggesting that it comes from just one experiment (The n numbers are not reported in the legend). Please repeat the experiment 3 times and provide quantification with error bars and statistical testing.

Response: We performed additional western blot analysis (total three times), replaced Fig. S2E (now Fig. S2G), and error bars and statistical validation results were included (bottom panel).

Fig. S3A, C-D. The quantifications do not have any error bars, suggesting that they come from just one experiment (The n numbers are not reported in the legends). Please repeat the experiments 3 times and provide proper quantification with error bars and statistical testing.

Response: We performed western blot analysis related to Fig. S3A, C-D for 3 times and statistical analysis. We replaced bar graphs with new ones with error bars and statistical validation. The conclusion was not changed.

Figure 3C suggests that the recovered H3K9me2 at 72h does not match with WT cells (considering the 9% variance explained by PC2), suggesting that either not all H3K9me2 recovers in this time frame or the recovery occurs in different locations. The conclusion that 'As H3K9me2 almost completely recovered three days after 174 UNC0642 removal' should be modified accordingly.

Response: As the reviewer pointed out, H3K9me2 profile was not completely recovered at 72h after UNC0642 removal from H3K9me2 ChIP-seq data (Fig. 3C).

However, H3K9me2 amount is almost completely recovered from western blotting data (Fig. S3A). The sentence the reviewer pointed out, 'H3K9me2 almost completely recovered three days after 174 UNC0642 removal', we just stated it based on the western blotting data. But, we deleted "completely" from the sentence and also changed other sentence about the PCA analysis to "Principal component analysis of H3K9me2 reads per kilobase million (RPKM) in the 80-kb genomic window showed that the H3K9me2 profile gradually approached that of WT after UNC0642 withdrawal, although had not completely recovered 72 h after withdrawal (Fig. 3C)." page 10, line 181-183.

Figure 4. Given the widespread occurrence of H3K9me2 across the genome (Fig. 1C) the reported numbers of upregulated genes are minimal, suggesting a limited effect of H3K9me2 on transcriptional regulation. The number of downregulated genes should also be reported.

Response: As we performed RNA-seq again as biological replicate, the number of differentially expressed genes was changed (Fig S4A). We described as follows, "From RNA-seq of WT and *Setdb1* KO mESCs treated with or without UNC0642, we identified 495, 194, and 79 upregulated genes and 61, 3, and 0 downregulated genes in UNC0642-treated *Setdb1* KO, *Setdb1* KO, and UNC0642-treated mESCs, respectively (Additional file 1: Fig. S4A). Only 3.5, 1.4, and 0.5% of genes that harbor G9a/GLP-independent H3K9me2 within 5 kb of TSSs were upregulated in these three mESCs, suggesting that G9a/GLP-independent H3K9me2 were mostly dispensable for transcriptional silencing." page 10, line 200 – page 11, line 206.

While the data shown in Figure 4A suggest that upregulated genes are often proximal to H3K9me2 domains this is not surprising given the prevalence of H3K9me2, even in UNC-treated cells (Fig. S1C). The authors should provide controls for this analysis. Are the upregulated genes more likely to be within 5KB to G9a/GLP-independent H3K9me2 domains than non-changed genes or downregulated genes? What % of G9a/GLP independent K9me2 domain genes (within 5KB of the TSS) are upregulated/unchanged/downregulated?

Response: We added a graph of total genes marked with G9a/GLP-independent H3K9me2 or H3K9me3 within 5kb from their TSS in the revised Fig. 4A. We did not include that of downregulated genes due to a small number of downregulated genes. Both G9a/GLP-independent H3K9me2 and H3K9me3 were more frequently associated with the upregulated genes than total genes. However, as stated above, only 0.5, 1.4 and 3.5 % of the genes among marked with G9a/GLP-independent H3K9me2 within 5kb from their TSS were upregulated in UNC, *Setdb1* KO and UNC-treated *Setdb1* KO mESCs, respectively.

The authors' statement that H3K9me2 correlated with gene repression more than H3K9me3 is too general and misleading. A more correct conclusion would be that H3K9me2 correlated more with gene repression of UNC-dependent genes than H3K9me3.

This is of course not surprising and exhibits circular reasoning. These genes were already shown to be regulated by H3K9me2 (upregulated upon UNC treatment), thus the fact that they have stronger enrichment of H3K9me2 than me3 is expected. Any comparison between the repressive effects of H3K9me2 and me3 cannot be conducted unless a proper analysis is performed removing K9me2 from K9me2-only regions and K9me3 from K9me3-only regions.

Response: We agree with the reviewer's comment "the authors' statement is too general and misleading". Therefore, we deleted this statement "Therefore, H3K9me2 correlated with gene repression to a greater extent than H3K9me3." from the revised manuscript.

Figure 4D. No statistics are presented for the expression plot, thus it is not possible to determine which group is significantly upregulated.

Response: We performed Tukey's test to Fig. 4D and all samples showed a statistically significant upregulation of genes marked with G9a/GLP-independent H3K9me2 and no H3K9me3 around their TSS.

In Figure 4E it appears that these genes do not possess H3K9me2 in control cells and only gain H3K9me2 in UNC treated cells. The H3K9me2 levels in control cells should also be plotted in Figure 4D. This also means that expression differences should be compared between *Setdb1* KO + UNC to UNC-treated controls in both Figure 4D and F, rather than to ?. It is not clear to which specific groups the authors refer when they suggest they observe upregulation, and most importantly, without statistics again it is impossible to conclude.

Response: We also plotted the H3K9me2 levels in WT cells and *Setdb1* KO mESCs in Figure 4D upper center panel. The lower H3K9me2 levels of control (WT) cells than UNC-treated ones might be due to a large difference of H3K9me2 coverage on genome between these two samples. The higher H3K9me2 levels around TSSs in WT than those in *Setdb1* KO mESCs as shown in new Fig. 4D indicates that H3K9me2 is already enriched around genes focused on in Fig. 4D in WT cells before UNC-treatment. We performed Tukey's test in Fig. 4D (left panel) and F (bottom RT-qPCR data) and showed a statistically significant upregulation of these genes in UNC-treated *Setdb1* KO mESCs compared to both WT or *Setdb1* KO cells and UNC-treated WT cells. In addition, we modified a sentence in page 12, line 235-238 to clarify which group was upregulated as follows: "Although these genes were not clearly marked with H3K9me3, they were upregulated in *Setdb1* KO mESCs both with or without UNC0642 (Fig. 4D, E). The upregulation of the three selected genes (*Nlrp4c*, *Gm4971*, and *Trpd52l3*) in UNC0642-treated *Setdb1* KO mESCs was validated using qRT-PCR (Fig. 4F)."

Furthermore, this suggests that the acquisition of H3K9me2 on these genes is an artifact of the inhibitor and thus not physiologically relevant. Can the authors provide evidence that G9a-GLP independent, *Setdb1*-dependent H3K9me2 actually occurs in wild-type cells? If not then analysis of the transcriptional effects of the loss of this modification must be

interpreted with caution. Do these regions also acquire H3K9me2 in G9a/GLP knockouts for example?

Response: We added the H3K9me2 levels of these regions in *G9a/GLP* DKO mESCs in new Fig. S4F. H3K9me2 in *G9a/GLP* DKO mESCs was also enriched in genes with H3K9me2 but without H3K9me3. Thus, the acquisition of H3K9me2 on these genes is not an artifact of the inhibitor. We described this on page 12, line 233-235.

“Furthermore, as shown in Fig. 4D, the H3K9me2 on these genes was lower in *Setdb1* KO mESCs than in control cells. Thus, G9a/GLP independent, SETDB1-dependent H3K9me2 is actually present in WT cells.”

Figure 5. The data demonstrate a clear correlation between H3K9me2 loss in UNC-treated cells and an increase in compartment score. As this is an important conclusion this analysis should be extended. First in order to validate this observation, Hi-C data should be generated from G9a/GLP DKO ES cells to determine if a similar trend is observed. Secondly, the authors mention that increases in compartment scores are observed in both A and B compartments, but the data is not presented. Please add this analysis to the Figure. Thirdly, they should correlate compartment scores to H3K9me2 domains. In UNC treated cells domains are smaller. How does this correlate to compartment scores? Do compartments also shrink in size? Do strongly affected H3K9me2 domains result in compartment switching or more pronounced differences in compartment scores? Do the few switching compartment lose H3K9me2? Finally the authors should present Hi-C plots of a representative chromosome/genomic region along with associated H3K9me2 profiles to show compartment weakening, correlating to H3K9me2 loss.

Response: We have responded to reviewer’s requests related to Fig. 5 as follows.

- Hi-C analysis of *G9a/GLP* DKO mESCs

We reanalyzed the recently published Hi-C data from *G9a* KO mESCs deposited in GSE138102 (Jiang Q et al., Cell reports, 2020). UNC-treated cells and *G9a/GLP* DKO mESCs showed similar patterns in H3K9me2 and compartment score change as shown in new Fig. S5B-F.

Secondly, the authors mention that increases in compartment scores are observed in both A and B compartments, but the data is not presented.

Response: We mentioned this statement specific to upregulated genes in UNC-treated *Setdb1* KO mESCs shown in old Fig. 5F. To make this statement more clearly, we modified the sentence in page 13, line 266-267 as follows, “The genes upregulated in UNC0642-treated WT or *Setdb1* KO mESCs showed a significant increase in compartment scores (Fig. 5H).”

Thirdly, they should correlate compartment scores to H3K9me2 domains. In UNC treated cells, domains are smaller. How does this correlate to compartment scores?

Response: Correlation between the change of fraction of H3K9me2 domains in each 100-kb windows and the difference in compartment score were analyzed in new Fig.

5D-E and new Fig. S5C-D. These results shows that increased compartment score induced by UNC0642 treatment is well correlated with decreased H3K9me2 domain size.

- Do strongly affected H3K9me2 domains result in compartment switching or more pronounced differences in compartment scores? Do the few switching compartment lose H3K9me2?

Response: New Fig. 5F-G and new Fig. S5E-F shows that “B to A” conversion was correlated with decreased size of H3K9me2 domains and the regions with decreased H3K9me2 domains was frequently overlapped with “B to A” conversion.

- Finally the authors should present Hi-C plots of a representative chromosome/genomic region along with associated H3K9me2 profiles to show compartment weakening, correlating to H3K9me2 loss.

Response: We added representative genomic region in new Fig. S5G, which showed correlation of decreased H3K9me2 with increased compartment score.

Minor comments

Reference 39: Invalid citation

Response: That was an error from EndNote software and we corrected that and now ref 39 is

39. Durand NC, Shamim MS, Machol I, Rao SS, Huntley MH, Lander ES, Aiden EL: Juicer Provides a One-Click System for Analyzing Loop-Resolution Hi-C Experiments. *Cell Syst* 2016, 3:95-98.

Fig. S1A Ponso should be Ponceau

Response: We corrected it.

Figure 1b. An increase rather than a decrease in the H3K9me2 signal is observed in compartment A in *Setdb1* KO and in compartment B in *Suv39h1/2* DKO. Is the y axis labelled incorrectly? Also applies to Figure 1F.

Response: As the y axis represents a difference of the H3K9me2 signal between *Setdb1* KO mESCs and WT mESCs (“*Setdb1* KO” – “WT”), a negative value of y axis indicates a decrease of H3K9me2 signal in *Setdb1* KO mESCs. Thus, label of the y axis is correct.

Fig. S1C please provide a quantification of H3K9me2 levels from 3 independent experiments

Response: We performed 3 independent experiments in all H3K9me2 or 3 western blotting analyses (Fig. S1C, S1M, S1N, S2F, S2G, S3A, S3C and S3D), and provided quantification data with statistical validation.

Fig. S1D please report the number of times the experiment was repeated.

Response: We performed IF analysis at least 2 times and described about this in the legend of Fig. S1.

Fig. S1H The legend is missing.

Response: We corrected it.

Fig. S1L and M please provide quantification of H3K9me2 and 3.

Response: We provided quantification data for them (the revised Fig. S1M and N).

Reviewer #2 (Remarks to the Author):

The paper by Fukuda et al. examines the role of H3K9me2 and H3K9me3 in gene regulation and chromatin conformation in mouse ESCs and iMEFs. While multiple previous studies have addressed some of these questions the authors try to unravel the relative contribution of five H3K9 methyltransferases in putting down H3K9me2 and silencing genes. By using multiple genetic mutants in combination with a specific G9a inhibitor the authors claim that there is significant amount of G9a/GLP-independent H3K9me2. This is mainly dependent on Setdb1 and to a lesser extent Suv39h1/2. This G9a/GLP-independent H3K9me2 correlates with gene repression and regulates chromatin compartmentalization. These claims are novel and likely of interest for the readers of Communication Biology. Indeed, by performing meticulous chromatin based assays in a panel of mutants the authors uncover the importance of G9a/GLP-independent H3K9me2. However in my opinion there are some important weaknesses to this manuscript and the main conclusions are to some extent misleading.

Nevertheless I feel that this is an elegant and thorough study, after addressing my comments I support the publication of this manuscript. Below I'm attaching my detailed criticism.

Response: First of all, thank you for your quite positive evaluation on our work.

Major points:

-In general the experiments are well designed. However I feel like the data analysis is sometimes misleading. It is clear from the correlations that in the vast majority of G9a-independent H3K9me2 does not regulate gene expression or chromatin conformation. This mark is very wide-spread and yet very few genes become derepressed. This was also reported for G9a-dependent H3K9me2 (Zylicz et al, 2015). Please provide quantitative

data on what fraction of G9a-independent H3K9me2 targets are actually regulated by this mark?

Response: We added the sentence below in page 10, line 203- page 11, line 206. “Only 3.5, 1.4, and 0.5% of genes that harbor G9a/GLP-independent H3K9me2 within 5 kb of TSSs were upregulated in these three mESCs, suggesting that G9a/GLP-independent H3K9me2 were mostly dispensable for transcriptional silencing.”

The authors never show how many genes are downregulated in all the mutants, this data has to be shown.

Response: We described the number of downregulated genes in page 10, line 201-203 as below. “we identified 495, 194, and 79 upregulated genes and 61, 3, and 0 downregulated genes in UNC0642-treated *Setdb1* KO, *Setdb1* KO, and UNC0642-treated mESCs, respectively (Additional file 1: Fig. S4A).”

The authors use constitutive KOs which will accumulate secondary gene expression changes, this should at least be discussed. The authors should provide a clear conclusion acknowledging that for most targets H3K9me2 does not seem to be functional.

Response: As we used conditional *Setdb1* KO mESCs for RNA-seq analysis, we do not think such secondary gene expression changes are large. We described on page 11, line 205-206 that “G9a/GLP-independent H3K9me2 were mostly dispensable for transcriptional silencing.”

RNAseq analysis throughout the paper constantly uses different thresholds for Fold Change (e.g. Figure s4b; Figure s4c; and in page 10 line 222). Why is this?

Response: We used a stringent threshold to identify differentially expressed genes in Fig. S4A, B (FDR<0.05, FC>10). As we just wanted to know whether upregulated genes in UNC0642-treated WT mESCs, which was identified by the threshold mentioned above, could be also derepressed in *Setdb1* KO mESCs, we used a weaker criterion to investigate it in Fig. S4C. Then, we described as follows. “Although only 27 genes were upregulated in both UNC0642-treated mESCs and *Setdb1* KO mESCs, among the upregulated genes in UNC0642-treated mESCs, 16 of 19 and 56 of 59 genes located in the A and B compartments, respectively, were derepressed in *Setdb1* KO mESCs at least twice. Therefore, most genes repressed by G9a/GLP-dependent H3K9me2 were also potentially regulated by SETDB1 (Additional file 1: Fig. S4C). page 11, line 210-215

It is vital to have at least duplicates of RNAseq experiments to perform robust analysis.

Response: We performed duplicate of RNA-seq analysis and combined the duplicate to identify differentially expressed genes. We replaced Fig. 4A-E, Fig. S4A-E with data from combined data. Although the number of differentially expressed genes was changed, the conclusion was not affected.

Figure 4: in this analysis the authors claim that H3K9me2 correlates better with gene repression. However the analysis is missing key controls. How will the analysis look at random genes throughout the genome?

Response: We added the data from total genes in Fig. 4A. Both G9a/GLP-independent H3K9me2 and H3K9me3 were more frequently associated with upregulated genes than total genes.

H3K9me2 is much more prevalent than H3K9me3, are the correlations observed actually significant when compared to remaining parts of the genome?

Response: G9a/GLP-independent H3K9me2 is more abundant in the genome than H3K9me3 as below (only for the reviewer).

The authors claim that H3K9me2 regulates chromatin compartmentalisation however the observed changes are relatively weak. Can they be fully explained by transcriptional upregulation? Are compartmentalisation scores affected also on H3K9me2 targets that do not get upregulated in mutants? If not, than it seems unlikely that H3K9me2 actually regulates chromatin compartments, it is rather affected by gene expression. To strengthen this point the authors should do HiC in TKO cells treated with UNC and see if the conformation is more affected.

Response: This is an important point. The number of genes in each 100-kb bin which showed both an increased compartment score and a decreased H3K9me2 in UNC-treated mESCs was analyzed, and the analysis revealed that about half of these 100-kb bin does not contain genes as below (only for the reviewer). Thus, gene activation cannot fully explain an increased compartment score in UNC-treated mESCs, and we decided not to analyze 3D genome organization in TKO mESCs treated with UNC in the current paper.

Conclusion on p14 l.298-299 is not supported by the data and has to be rewritten. There is no convincing data that H3K9me2 is one of the key players of determinant mechanism.

Response: We modified the sentence as the following,

Original: Therefore, we propose that H3K9me2 contributes to heterochromatin compartment formation as one of the key players of determinant mechanisms (Figure 6 bottom).

Revised one: Therefore, we hypothesize that H3K9 methyltransferases play a role in maintaining the genome in a more inactive compartment setting by histone methylation (Figure 6, bottom). Page 17, line 327-329.

-The authors do not actually analyse H3K9me2 spreading upon UNC removal. They convincingly uncover that regions with more G9a-independent H3K9me2 recover faster but do not look at spreading from these regions. This analysis should be performed, how do windows directly bordering G9a-independent domains behave? Also the use of UNC, a catalytic inhibitor, implies that G9a/GLP can still play a function in quick re-establishing of H3K9me2. Indeed, it is possible that G9a/GLP is inactive but pre-bound to regions which recover faster upon UNC removal. This possibility should be discussed. Conclusion on spreading (e.g. p13 l279) needs to be amended.

Response: We analyzed the recovery of H3K9me2 around G9a/GLP-independent H3K9me2 in new Fig. S3E and that did not show H3K9me2 spreading from G9a/GLP-independent H3K9me2. Thus, we could not get a crucial evidence of H3K9me2 spreading from G9a/GLP-independent H3K9me2. We described about the limitation of H3K9me2 spreading and the possibility of G9a/GLP bindings to G9a/GLP-independent H3K9me2 to recover H3K9me2 efficiently on p16, line 304-307 as follows “Thus, G9a/GLP-independent H3K9me2 facilitates efficient H3K9me2 recovery, but its spreading might be limited. The efficient recovery of genomic regions marked with G9a/GLP-independent H3K9me2 might be supported by ANK-domain-mediated pre-bound G9a/GLP to H3K9 methylated nucleosomes upon UNC0642 removal.”

Conclusion from p6 l126-7 claims a big contribution of Suv39h1/2 in mediating H3K9me2 however the ChIPseq data did not reveal this (Fig 1H). As authors suggest this is because the repetitive fraction of the genome could be mainly affected. As the data stands the conclusions are not supported by the analysis. Authors should include H3K9me2 ChIPseq analysis of the repetitive fraction of the genome. Alternatively the conclusions should be significantly softened.

Response: We changed the sentence on p6 line 131-132 more softened as follows. “Therefore, G9a/GLP-independent H3K9me2 in the B compartments is mediated by both SETDB1 and SUV39H1/2.”. Page 7, line 132-133.

-Statistical analysis is missing throughout the manuscript especially in multiple analysis of genome-wide datasets.

Response: Statistical analysis was performed for genome-wide datasets (Fig. 3EF, 4AD and Fig. 5C-F). We added correlation data between biological replicates for genome-wide datasets in new Supplementary Table S4. In addition, we showed reproducibility of compartment dependent regulation of H3K9me2 by SETDB1 and SUV39H1/2 in new Fig. S1I, Fig. S2B, D. We also showed reproducibility of correlation of increased compartment score with decreased H3K9me2 in new Fig. S5B-F.

-The title is not informative and does not provide any insight into the conclusions of this paper.

Response: We revised the title as “Regulation of mammalian 3D genome organization and histone H3K9 dimethylation by H3K9 methyltransferases”. Hope new title is acceptable although we know it is still not perfect one.

Minor comments:

-Model in Figure 6 proposes that Setdb1KO+Unc treatment results in relocation of genes away from nuclear lamina- this has not been addressed in this paper.

Response: Nuclear lamina was removed from Fig. 6.

-Figure S4C- what are the diagonal lines representing?

Response: The lines probably emerged during conversion process from PowerPoint file to pdf file at electric submission. So, no specific means.

-Immunoblot staining in e.g. Fig 1 is not spelled Ponso but Ponceau S

Response: We corrected it.

-H3K9me2 genome wide pattern is better visualised when windows are shown as $\text{Log}_2(\text{H3K9me2}/\text{Input})$, this was done in many publications e.g. Lienert et al, 2011.

Response: We added the track H3K9me2 profiles normalized by Input to Fig. 1C and Fig. 2A.

-Nora et al, 2012 should be cited along Dixon et al for TADs.

Response: We cited this reference on Page 4, line 60.

Reviewer #3 (Remarks to the Author):

In "Functional correlation of H3K9me2 and nuclear compartment formation" Fukuda and colleagues map H3K9me2 in mESC and MEF cells genome wide to determine which methyltransferases regulate deposition of H3K9me2 in different genomic compartments. The authors determine that Setdb1 (thought to be primarily an H3K9me3 methyltransferase) is required for methylation of H3K9me2 in the A compartment of the genome. The authors performed Hi-C to map changes in the compartments cause by loss and/or inhibition of the 5 methyltransferases and find that largely the compartments remain unchanged. The authors' work attempts to address a fundamental question in chromatin biology of how histone modifications regulate the establishment and maintenance of 3D genome architecture, an important question in the field. However, as currently constructed the manuscript fall short of providing evidence for this concept.

The manuscript title "Functional correlation of H3K9me2 and nuclear compartment formation" Is not supported by the authors own data in figure 5, where ~95% of chromatin Hi-C domains do not change when the authors manipulate the H3K9me2 methyltransferases. The authors do not demonstrate formation of any compartments is dependent on H3K9me2 or any of the methyltransferases. While the observations of which methyltransferases are responsible for methylation at different genomic regions is important, this does not satisfy the authors title and goals.

Response: We revised title as "Regulation of mammalian 3D genome organization and histone H3K9 dimethylation by H3K9 methyltransferases". Hope new title is acceptable although we know it is still not perfect one.

While the authors do provide convincing evidence that Setdb1 is required for some H3K9me2 deposition, the authors do not provide evidence that the enzyme is directly depositing H3K9me2. The authors do not consider the role of the demethylases in the formation of the H3K9me2 domains. The experiments in Suv39H DKO and SETDB1 KO cells yielding reduced H3K9me2 domains could be because there is reduced H3K9me3 there is less target for demethylation to H3K9me2. Rather than the methyltransferases producing H3K9me2 directly. The authors should, at a minimum, moderate their language throughout to reflect this. However, addressing if the demethylases are important would significantly strengthen the manuscript.

Response: Yes, such demethylation-mediated indirect production of H3K9me2 from H3K9me3 deposited by SETDB1 or SUV39H is also possible. We described about H3K9 demethylase in discussion section on page 17, line 321-324 as follows. "Another possible mechanism to generate G9a/GLP-independent H3K9me2 is demethylation of H3K9me3 by H3K9 demethylase such as KDM4A [37], and enrichment of SETDB1-dependent H3K9me2 without H3K9me3 downstream of TTSS of the SETDB1-target genes (Fig. 4B and C, left panel) might be caused by demethylation of H3K9me3."

The authors need to provide validation (either their own or published for the monoclonal H3K9me2 and H3K9me3 antibodies used for ChIP) that they really are specific for

dimethyl or trimethyl modifications as this has historically been a big problem on the cross reactivity of these antibodies.

Response: The specificity of antibodies for H3K9me2 (6D11) and H3K9me3 (2F3) used ChIP was reported in Hayashi-Takanaka et al., NAR, 2011 and Chandra et al., Mol. Cell, 2012. We described that in Antibodies, Materials and Methods section.

Minor Points:

Many figures need to have improved labeling throughout. Many graphs are lack statistical analysis. The methods need to be improved with more details for the sequencing approaches. In particular, how many replicates were performed and how were the replicates combined to produce the tracks presented. Many of the sequencing tracks need to have higher resolution images as well.

Response: We responded to those comments (Fig. labeling, statistics, detailed seq method and so on), and replaced sequencing tracks with higher resolution images.

The sentence on lines 79-81 is very confusing and I had to read it 3 times to understand what the authors meant when compared to the data. My initial reaction reading the sentence and looking at the figure was that the authors were wrong, but after carefully rereading 5 times I understood the intent. Please clarify the writing.

Response: We revised the sentence as the following. "H3K9me2 profiles in *Setdb1* KO and *Suv39h1/2* DKO mESCs were highly correlated with those of wild-type (WT) mESCs (Additional file 1: Fig. S1B). However, we found decreased H3K9me2 in the A compartments of *Setdb1* KO mESCs and in the B compartments of *Suv39h1/2* DKO mESCs, respectively (Fig. 1B)." Page 5, line 79-82. Hope this is better to understand what we want to say.

Please include error bars on all bar graphs of western blot quantification and ensure that all blots have quantification.

Response: We added error bars and quantification data to all western blotting data.

Figure 1D- please present this as a box and whisker or violin plot to clearly indicate the median and quartiles. Is this statistically different than wild type cells?

Response: The Y-axis of Fig. 1D represents the relative total length of H3K9me2 domain in *Setdb1* KO or *Suv39h1/2* DKO mESCs to WT mESCs as shown in Fig. 1A-C. Thus, we could not make a box and whisker or violin plot for this data. Only for information to the reviewer, following is the distribution of H3K9me2 domain length in the A/B compartments between WT and KO mESCs.

Figure 1E- the authors should analyze the cells by flow cytometry to ensure the cells are in equal distributions of the cell cycle. If the UNC treated cells arrest (or the G9a/GLP DKO) cells in a different phase this could explain the clustering and domain differences.

Response: We performed cell cycle analysis in WT and G9a/GLP DKO mESCs by Hoechst staining followed by FACS analysis 3 times. These two cell lines showed similar cell cycle pattern as bellow (only for the reviewer).

Figure 1H, please include in the figure a label along the X axis to indicate what the heatmaps are in relation to.

Response: We modified Fig. 1H by adding a label along the X axis “-10kb, 5’, 3’ and 10kb”.

Figure 2B- Please use box and whisker or violin plots to plot the distribution to provide more information than the basic bar plots.

Response: A Violin plot of length distribution of H3K9me2 domains was generated as below. However, as it is hard to see majority of H3K9me2 domains due to relatively short length of them, we did not include this figure in the manuscript (only for the reviewer).

Figure 3C- “Principal component analysis of H3K9me2 Reads Per Kilobase Million (RPKM) in the 80-kb genomic window showed gradual H3K9me2 recovery after UNC0642 removal (Fig. 3C)” This is not true, this simply says the timepoints have different principal component values. It says nothing about recovery of H3K9me2. If the authors want to make this point, they should plot a metagene analysis centered on the H3K9me2 peaks at time point 0 and plot average H3K9me2 levels up and downstream from this point (maybe +/-100 KB? Or region at authors discretion) at each of the subsequent time points.

Response: Thank you for reasonable comment. We omitted the recovery issue and modified the sentence on page 10, line 181-183 as follows. “Principal component analysis of H3K9me2 reads per kilobase million (RPKM) in the 80-kb genomic window showed that the H3K9me2 profile gradually approached that of WT after UNC0642 withdrawal, although had not completely recovered 72 h after withdrawal (Fig. 3C).”

Line 237- “correlation was mild” No correlation coefficient by any correlation measure was provided in figure 5. To make this statement please provide a proper correlation analysis. Was the analysis in Figure 5C-D only performed on compartments with changes

(i.e only 5% of compartments?) Figure 5 seems vastly overinterpreted, the simplest conclusion is that H3K9me2 did not affect compartment formation.

Response: Again, thank you for this comment. We removed the phrase “correlation was mild” from the manuscript. We did not use only regions showing compartment change but all genomic regions for correlation analysis of H3K9me2 and compartment score. We revised about overall maintenance of compartment pattern in H3K9me2 depleted cells on page 13, line 267-270 as follows. “Therefore, decreased H3K9me2 correlated with the relocation of the target genes to more active compartments; moreover, this relocation is correlated with transcriptional activation, although the overall compartment pattern is maintained in H3K9me2-depleted cells (Fig. 5l).”

Fig S1C- please provide quantification of the amount of reduction. It's not clear what Pan-H3 is for an antibody, is this acetylated H3 or just recognizing all H3 in which case just label it H3. If it is supposed to be the loading control, then the UNC treated samples are clearly reduced in histones and another loading control should be chosen.

Response: We provided quantification data for all western blotting data. Pan-H3 represents all histone H3. We replaced “Pan-H3” to “Histone H3” in all figures. Among the triplicate experiments, amount of H3 was not significantly different (per same amount of loaded sample) which is shown in below (only for the reviewer). Furthermore, H3 is not used for the loading control. For comparison among the samples (wt, KO and UNC0642-treated cells), we needed the value of H3 methylation signals per same amount of H3. Therefore, it is not critical much even if amount of H3 is affected by the UNC0642 treatment.

Western blotting of histone H3 in WT and UNC0642 treated mESCs. Data are mean ± SEM; n = 3. NS: P>0.05. P-value was calculated by paired Student's t-test.

Figure S1D- please include a scale bar on all micrograph images.

Response: We included a scale bar on all micrograph images.

Figure S1F is blurry and a low-resolution screenshot from IGV. Please also clearly indicate the signal and the domain tracks as was done in the main figures.

Response: We replaced the Fig. S1F with a high-resolution image.

Figure S2D spell Ponceau correctly. Why does H3K9me3 increase in the *Setdb1* ko cells?

Response: We corrected the spell. The increase of H3K9me3 in the *Setdb1* KO cells was not significant as shown in new Fig. S2F.

Figure S4- what do the lines emanating from the corner of the violin plots mean?

Response: The lines probably emerged during conversion process from PowerPoint file to pdf file at electric submission. So, no specific means.

REVIEWERS' COMMENTS:

Reviewer #1 (Remarks to the Author):

I am satisfied with the changes the authors have made to the manuscript and I now feel that my comments have been addressed in full. Thus I am happy to recommend publication of this manuscript in Communications Biology.

Reviewer #2 (Remarks to the Author):

The revised manuscript addresses most of my concerns even though the authors decided not to perform HiC in TKO Unc treated cells. I am still convinced that the Figure 4A needs restructuring as the comparisons are not meaningful. Once this is addressed I support the publication of this manuscript.

Detailed criticism:

*Fig4A I think the comparison of H3K9me2 to H3K9me3 is meaningless since H3K9me3 is much less frequent in the genome than H3K9me2. This is even shown by the authors in the response document and now visible from the control of all genes. You can only compare blue bars to other blue bars and orange ones to other orange bars. Please restructure this figure. This will show that both H3K9me2 and me3 are on average enriched in the upregulated genes when compared to all other genes.

*Fig4A Please include the title for the Y axis

*The sentences on p17 line 327-29 and page 11, line 210-215 are grammatically clunky

Reviewer #3 (Remarks to the Author):

The authors have addressed my concerns. The sequencing data and replicates throughout are now adequately described. The language changes made by the authors have solved the interpretation issues throughout. I support publication of the article in Communications Biology.

Reviewers' comments:

Reviewer #2 (Remarks to the Author):

The revised manuscript addresses most of my concerns even though the authors decided not to perform HiC in TKO Unc treated cells. I am still convinced that the Figure 4A needs restructuring as the comparisons are not meaningful. Once this is addressed I support the publication of this manuscript.

RESPONSE: First of all, thank you for your additional efforts to re-evaluate our revised manuscript.

Detailed criticism:

*Fig4A I think the comparison of H3K9me2 to H3K9me3 is meaningless since H3K9me3 is much less frequent in the genome than H3K9me2. This is even shown by the authors in the response document and now visible from the control of all genes. You can only compare blue bars to other blue bars and orange ones to other orange bars. Please restructure this figure. This will show that both H3K9me2 and me3 are on average enriched in the upregulated genes when compared to all other genes.

RESPONSE: Following to the reviewer's suggestion, the graph of H3K9me2 and H3K9me3 in Fig.4A was split, and we modified the sentence which refers comparing enrichment of H3K9me2 around TSS with H3K9me3 as follows. "Both G9a/GLP-independent H3K9me2 and H3K9me3 were frequently found within 5 kb from the transcriptional start sites (TSSs) of upregulated genes in each condition (Fig. 4a)." p11, line 228- p12, line 230.

*Fig4A Please include the title for the Y axis

RESPONSE: We added the title for the Y axis of revised Fig. 4a as "Fraction of genes with G9a/GLP-independent H3K9me2 or H3K9me3 (%)"

*The sentences on p17 line 327-29 and page 11, line 210-215 are grammatically clunky

RESPONSE: We revised these sentences as follows.

P17 line 327-329

Original: Therefore, we hypothesize that H3K9 methyltransferases play a role in maintaining the genome in a more inactive compartment setting by histone

methylation (Figure 6, bottom).

Revised: Therefore, we propose that H3K9 dimethylation contributes to the inactive compartment setting in mammals (Fig. 6c). P17, line 331-332.

P11, line 210-215

Original: Although only 27 genes were upregulated in both UNC0642-treated mESCs and *Setdb1* KO mESCs, among the upregulated genes in UNC0642-treated mESCs, 16 of 19 and 56 of 59 genes located in the A and B compartments, respectively, were derepressed in *Setdb1* KO mESCs at least twice. Therefore, most genes repressed by G9a/GLP-dependent H3K9me2 were also potentially regulated by SETDB1 (Additional file 1: Fig. S4C).

Revised: Only 27 genes were commonly upregulated in UNC0642-treated mESCs and *Setdb1* KO mESCs (Supplementary Fig. 4a). However, among the upregulated genes in UNC0642-treated mESCs ($2+50+27=79$ in Supplementary Fig. 4a), 72 genes (16 and 56 in the A and B compartment, respectively) were derepressed in *Setdb1* KO mESCs at least twice (Supplementary Fig. 4c), suggesting that most genes repressed by G9a/GLP-dependent H3K9me2 are also regulated by SETDB1. P11, line 213-218.